# Stable Matching with Ties: Approximation Ratios and Learning

**Shiyun Lin**[*]
Center for Statistical Science
School of Mathematical Sciences, Peking University
shiyunlin@stu.pku.edu.cn

**Simon Mauras**
INRIA, FairPlay Joint Team
simon.mauras@inria.fr

**Nadav Merlis**
Technion - Israel Institute of Technology
nmerlis@technion.ac.il

**Vianney Perchet**
CREST, ENSAE, IP Paris
Criteo AI Lab, FairPlay Joint Team
Vianney.perchet@normalesup.org

## Abstract

We study matching markets with ties, where workers on one side of the market may have tied preferences over jobs, determined by their matching utilities. Unlike classical two-sided markets with strict preferences, no single stable matching exists that is utility-maximizing for all workers. To address this challenge, we introduce the *Optimal Stable Share* (OSS)-ratio, which measures the ratio of a worker's maximum achievable utility in any stable matching to their utility in a given matching. We prove that distributions over only stable matchings can incur linear utility losses, i.e., an $\Omega(N)$ OSS-ratio, where $N$ is the number of workers. To overcome this, we design an algorithm that efficiently computes a distribution over (possibly non-stable) matchings, achieving an asymptotically tight $\mathcal{O}(\log N)$ OSS-ratio. When exact utilities are unknown, our second algorithm guarantees workers a logarithmic approximation of their optimal utility under bounded instability. Finally, we extend our offline approximation results to a bandit learning setting where utilities are only observed for matched pairs. In this setting, we consider worker-optimal stable regret, design an adaptive algorithm that smoothly interpolates between markets with strict preferences and those with statistical ties, and establish a lower bound revealing the fundamental trade-off between strict and tied preference regimes.

## 1 Introduction

Two-sided matching markets are prevalent in various contexts, such as matching students to schools [2, 3], doctors to hospitals [50], or workers to jobs [5]. In this paper, we model the market as a *company* that assigns *jobs* to *workers*. Each participant has a preference ordering over the other side of the market. For example, jobs rank workers by ability, while workers rank jobs by personal preference. Stability ensures a fair equilibrium where workers receive sufficiently desirable jobs while respecting the preferences and priorities of all parties. When preferences are strict, the deferred acceptance algorithm [23] efficiently computes a worker-optimal stable matching – no worker can get a better job without violating stability.

In online marketplaces, for example, the online crowd-sourcing platform Amazon Mechanical Turk, workers are usually uncertain of their preferences over jobs at the beginning, since they do not have hands-on experience. However, there are numerous similar tasks to be delegated

---

[*]This work was performed when Shiyun Lin was a visiting student at CREST, ENSAE, IP Paris.

39th Conference on Neural Information Processing Systems (NeurIPS 2025).

on the platform and, fortunately, the uncertain preferences can thus be learnt during the iterative matchings. Recent research has explored this scenario within the framework of multi-player multi-armed bandits [42, 43, 8, 37]. Under the strict preferences assumption, these works combine bandit learning algorithms with the deferred acceptance procedure to guide the market toward the worker-optimal stable matching.

However, in real-life scenarios, workers could be indifferent between some jobs due to inherent uncertainty or coarse evaluations. For instance, conference management systems like the Toronto Paper Matching System (TPMS): while the system generates continuous scores to evaluate the suitability of each reviewer for a paper, which theoretically avoids ties, the bidding process introduces unavoidable indifference through discrete categorical ratings (e.g., "Eager", "Willing", "In a pinch", "Not willing"), creating *natural ties* in preferences. The challenge becomes even more pronounced in learning-based matching markets, where statistically indistinguishable utility estimates produce *effective ties* between options. This presents a fundamental limitation for bandit learning approaches, as standard algorithms typically fail to provide meaningful regret guarantees when facing such indifference structures in the preference landscape. In particular, when utility differences become small (statistically indistinguishable), existing regret bounds break down completely, and handling this regime was previously considered impossible [42].

With indifferent preferences, a stable matching can be obtained by arbitrarily breaking ties and applying the deferred-acceptance algorithm. However, the resulting matching is no longer worker-optimal, as different tie-breaking rules may lead to different stable matchings preferred by different workers – potentially creating dramatic utility disparities across outcomes. This challenge is particularly acute in bandit learning settings, where statistically indistinguishable utilities for one worker may lead to arbitrarily large regret for others due to the cascading effects of tie-breaking decisions. In fair resource allocation, fractional matching is a standard technique for balancing competing interests when a single integral matching is infeasible [33, 27, 9]. The Birkhoff-von Neumann (BvN) theorem [10, 58] establishes that such a fractional matching is equivalent to a probability distribution over integral matchings.

These observations motivate our core research question: *For markets with tied preferences, can we approximate a stable solution by considering distributions over matchings, while guaranteeing all workers a fair, minimum level of satisfaction?*

To answer this question, we define a worker's *optimal-stable-share* (OSS) as her maximum achievable utility across all stable matchings. We then introduce the *OSS-ratio* as a fairness metric, which measures the fraction of the OSS that each worker is guaranteed to receive under any allocation.

We begin by analyzing the offline setting with known preferences, establishing tight OSS-ratio bounds across different matching classes. These results naturally extend to settings with preference uncertainty. Building on these offline results, we further formulate the problem within a multi-player multi-armed bandit framework for online learning scenarios, and show how our approximation guarantees provide the crucial foundation for achieving sublinear regret in matching markets with indifference.

## 1.1 Main Contributions

**Offline Approximation Oracle and Matched Upper and Lower Bounds.** We first demonstrate that restricting to stable matchings yields only a trivial (and tight) lower bound on the OSS-ratio (Theorem 1), motivating our study for broader matching classes. We then establish a logarithmic lower bound for general matchings (Theorem 2) and construct an approximation oracle (Algorithm 1) achieving this bound while maintaining internal stability (Theorem 3).

**Robustness to Approximated Preferences.** We prove our positive results are robust to utility uncertainty: when exact utilities are unknown but lie within a given uncertainty set, we maintain the same guarantees with only an additive error bounded by the maximum uncertainty (Theorem 6). This holds especially for rectangular uncertainty sets, which model utility matrices estimated from data.

**Bandit Learning in Matching Markets with Indifference.** Building on our offline approximation results, we introduce $\alpha$-approximation stable regret $Reg_i^\alpha(T)$, using an $\alpha$-fraction of the optimal-stable-share as a tractable benchmark for markets with (statistical) ties. Our adaptive algorithm ETCO (Algorithm 3) seamlessly handles both strict and tied preferences. Theorem 7 establishes its regret bounds, which match the lower bound [52] in markets with large preference gaps. Theorem 8 further reveals a fundamental trade-off: no algorithm can simultaneously achieve optimal regret in both large-gap (standard regret) and small/no-gap (approximation regret) regimes.

## 1.2 Techniques Involved and Developed

The upper bound on the approximation ratio is the first key technical contribution of our paper. We establish this result via three main steps: 1) Introducing a novel component – the duplication index – into the algorithm design; 2) Constructing a directed forest where edges encode conflicts between workers competing for the same job copies across different matchings; 3) Leveraging the tree structure and stability constraints to derive the upper bound inductively.

In the bandit learning setting, the primary technical challenge and key contribution lie in the lower bound proof. To establish this result, we carefully construct two instances with 4 workers and 4 jobs, where the utility matrices differ in only one critical entry that determines whether meaningful ties exist. This construction reveals how ties in one worker's preferences propagate to affect other workers' regret. Furthermore, we employ an information-theoretic argument to demonstrate that the algorithm must sample this critical entry sufficiently often to avoid incurring linear regret. To our knowledge, we are the first to provably show a tradeoff between standard regret and approximation regret in bandit settings.

## 1.3 Related Work

**Stable Matching with Ties.** A natural extension of Gale and Shapley's work [23] considers settings with tied or incomplete preferences. Irving [29] introduced three stability notions - weak, strong, and super-stability - with weak stability being the most studies [25, 26, 35, 46], as it always guarantees existence, unlike strong or super-stability. However, weakly stable matchings may vary in size, and finding a maximum one is NP-hard [30], while verifying weak stability is NP-complete [46]. Unlike prior work focused on maximizing matching size, we instead study fair job allocations, ensuring each worker receives a utility within a guaranteed fraction of their optimal stable matching, and we characterize the approximation ratio of such allocations.

**Fairness in Two-sided Matching.** Recent work has increasingly addressed fairness in two-sided markets. In fair division, Freeman et al. [21] introduces *double envy-freeness up to one match* (DEF1) and *double maximin share guarantee* (DMMS) for many-to-many matching, while Igarashi et al. [28] studies many-to-one matching, enforcing EF1 for one side while preserving stability. In machine learning, Karni et al. [32] incorporates *preference-informed individual fairness* (PIIF) [34], requiring allocations to satisfy individual fairness [18] while respecting preferences. Our work diverges by focusing on one-to-one markets, where standard notions like EF1 and MMS are inapplicable. We propose a novel share-based fairness concept (OSS-ratio) to measure workers' gains relative to their optimal-stable-share. Our algorithm returns a random matching that is ex-ante stable (no justified envy) and ex-post internally stable, achieving a best-of-both-worlds guarantee.

**Bandit Learning in Matching Markets.** Das and Kamenica [16] first formalized bandit problems in matching markets, with subsequent work [42, 43, 8, 52, 37] exploring this model. In this setting, players (with unknown utilities) and arms (with known preferences) form a two-sided market. *Player-optimal stable regret* [42] measures the utility difference between a player's outcome and their optimal stable match. Yet, existing results are limited to markets with strict preferences, as stable regret becomes linear and ill-defined when ties exist. Kong et al. [38] recently studied indifference cases, but their player-pessimistic regret benchmark cannot recover optimal stable matches in tie-free settings. Our work bridges this gap by: (1) establishing a tight logarithmic OSS-ratio for offline matching with ties, (2) introducing approximation regret as a tractable objective for tied markets, and (3) developing an adaptive algorithm that achieves optimal regret bounds in both tied and tie-free settings.

## 2 Preliminaries

We model the matching market as a *company* that assigns jobs to workers. There are $N$ workers, $\mathcal{W} = \{w_1, w_2, \cdots, w_N\}$ and $K$ jobs, $\mathcal{A} = \{a_1, a_2, \cdots, a_K\}$. The company assigns jobs to workers such that each job is assigned to at most one worker and each worker performs at most one job. The assignment is therefore a matching $\mu$. We shall use $\mu(w)$ to represent the allocated job to worker $w$, and $\mu(a)$ to denote the worker with job $a$. If a worker $w$ or a job $a$ remains unmatched, we will use the notation $\mu(w) = \perp$ or $\mu(a) = \perp$.

For every job, the company has a strict rating over the workers based on their expertise and ability on this job. Specifically, if $w \succ_a w'$, worker $w$ performs job $a$ strictly better than $w'$. On the other hand,

workers also have preferences over the jobs, and it is possible that a worker is indifferent among several jobs. The preferences of workers on jobs are represented through a utility matrix $U$, where $U(w, a) \in [0, 1]$ denotes the preference of worker $w$ on job $a$. If $U(w, a) > U(w, a')$, worker $w$ prefers job $a$ over $a'$, and $U(w, a) = U(w, a')$ implies that $w$ is indifferent between jobs $a$ and $a'$. For simplicity, we will assume that a worker $w$ will refuse to be matched with job $a$ if it has utility $U(w, a) = 0$; stated otherwise, either $U(w, \perp)$ is positive but infinitely small or $U(w, \perp) = 0$ and ties are broken in favor of $\perp$. As a consequence, a problem instance $(U, P_a)$ is defined by a utility matrix $U$ and a preference profile $P_a$ representing the preferences of jobs over workers.

Stability is a key concept in two-sided matching markets, which ensures there is no *justified envy* in the market, i.e., the only jobs a worker prefers over her own job are the ones that she is less suitable to face than the currently assigned worker. When preferences include ties, multiple stability notions arise, and we focus on *weak stability* [29]. A matching $\mu$ is weakly stable if no worker-job pair exists where both strictly prefer each other over their allocated partners:

**Definition 1** (Weak Stability). *A matching $\mu$ is* weakly stable *if there is no* blocking pair $(w, a)$ *such that $w \succ_a \mu(a)$ and $U(w, a) > U(w, \mu(w))$.*

If a matching is weakly stable, there exists a tie-breaking mechanism such that this matching is stable in the resulting instance with strict preferences. Conversely, any stable matching that is generated using a tie-breaking mechanism is also weakly stable in the original instance. Without causing ambiguity, we will refer to *weak stable* as *stable* for brevity. Furthermore, *internally stable matching* [44] refers to a matching where there are no blocking pairs when only considering the matched workers and jobs.

**Definition 2** (Internal Stability). *A matching $\mu$ is* internally stable *if there is no* internally blocking pair $(w, a)$ *such that 1) both $w$ and $a$ are matched in $\mu$, and 2) $w \succ_a \mu(a)$ and $U(w, a) > U(w, \mu(w))$.*

Given a problem instance, we define the following classes of matchings: $\mathcal{M} := \{\mu : \mu \text{ is a matching}\}$, $\mathcal{S} := \{\mu : \mu \text{ is a stable matching}\}$, and $\mathcal{I} := \{\mu : \mu \text{ is an internally stable matching}\}$.

In a matching market with ties, stable matchings are not unique, given different tie-breaking mechanisms. A job $a$ is a *valid stable match* of worker $w$ if there exists a stable matching that matches $w$ with $a$. We say $a$ is the *optimal stable match* of worker $w$ if it is the most preferred valid stable match, i.e., there exists a matching $\mu^* \in \mathcal{S}$ such that $\mu^*(w) = a$ and $U(w, \mu^*(w)) = \max_{\mu \in \mathcal{S}} U(w, \mu(w))$. We call $U(w, \mu^*(w))$ the *optimal stable share* (OSS) for worker $w$, denoted as $U^*(w)$.

The canonical results in two-sided matching markets are the Gale-Shapley theorem and algorithm (GS) [23], which guarantee both the existence of stable matchings and an efficient $\mathcal{O}(n^2)$ computation. The GS algorithm operates through an iterative proposal process. First, workers sequentially propose to their most preferred available jobs. Each job tentatively accepts its most preferred proposal and rejects others. After that, rejected workers continue proposing to their next preferences. The process terminates when no rejections occur, yielding a stable matching. In markets with strict preferences, GS produces a matching that is optimal for all proposers. However, when preferences contain ties, this optimality no longer holds uniformly.

**Example 1** (Stable matching with indifference). *Let $\mathcal{W} = \{w_1, w_2, w_3\}$ be workers and $\mathcal{A} = \{a_1, a_2\}$ be jobs with $w_1 \succ w_2 \succ w_3$ for all jobs. The utility matrix that encodes the preference of workers over jobs is given by:*
$$U = \begin{bmatrix} 1 & 1 \\ 1 & 0 \\ 0 & 1 \end{bmatrix}$$

*There are 2 stable matchings in this instance: $\mu_1 = \{(w_1, a_1), (w_3, a_2)\}$, $\mu_2 = \{(w_1, a_2), (w_2, a_1)\}$. There are 4 extra non-empty internally stable matchings, where exactly one worker is assigned a job of utility 1, and unmatched workers/jobs cannot be involved in blocking pairs. All workers have an OSS of 1. More precisely, $w_1$ receives utility $U^*(w_1) = U(w_1, a_1) = U(w_1, a_2) = 1$ in both stable matchings, $w_2$ receives utility $U^*(w_2) = U(w_2, a_1) = 1$ in $\mu_2$, and $w_3$ receives utility $U^*(w_3) = U(w_3, a_2) = 1$ in $\mu_1$.*

Example 1 demonstrates that different workers may achieve their optimal outcomes in different stable matchings. However, it is impossible to simultaneously guarantee all workers their OSS with a single matching (even non-stable). Based on this impossibility result, a natural question arises as to *whether an allocation exists such that every worker is at least satisfied at a certain level*. Formally, given a problem instance and a class of matchings $\mathcal{C}$, we are interested in the following *optimal stable share-ratio* (OSS-ratio):

$$R_{\mathcal{C}} := \min_{D \in \Delta(\mathcal{C})} \max_{w \in \mathcal{W}} \frac{U^*(w)}{U_D(w)}, \tag{1}$$

where $\Delta(\mathcal{C})$ is the set of distributions over $\mathcal{C}$ and $\boldsymbol{U}_D(w)$ is worker $w$'s expected utility given a distribution $D$, i.e., $\boldsymbol{U}_D(w) = \mathbb{E}_{\mu \sim D}[\boldsymbol{U}(w, \mu(w))]$. When we are constrained to the set of matchings, stable matchings and internally stable matchings, $R_{\mathcal{M}}$, $R_{\mathcal{S}}$ and $R_{\mathcal{I}}$ are defined accordingly.

The OSS-ratio adopts a worst-case perspective by taking the *maximum over workers*, ensuring every worker receives a fair share of their optimal stable utility. Formally, if $\max_U R_{\mathcal{M}} \leq \alpha$, then every worker $w_i$ is guaranteed at least $\frac{1}{\alpha}\boldsymbol{U}^*(w_i)$ in expectation, regardless of the market's preference structure. The *minimum over distributions* reflects a central planner's optimization: the distribution represents a rotating schedule (e.g., matchings in the support correspond to daily assignments), and *restricted support* encodes practical constraints. For instance, limiting support to internally stable matchings ensures no justified envy arises between co-present workers in any schedule realization.

# 3 Approximation Ratios for Stable Matching with Ties

In this section, we aim to characterize the scale of the OSS-ratio $R_{\mathcal{C}}$ from the worker's perspective, which allows for ties, while additional findings related to the job side are provided in Appendix J. As a first observation, $\mathcal{S} \subset \mathcal{I} \subset \mathcal{M}$ implies $R_{\mathcal{M}} \leq R_{\mathcal{I}} \leq R_{\mathcal{S}}$, and $R_{\mathcal{S}} \leq N$, since uniformly selecting a worker and their favored stable matching achieves this bound.

## 3.1 Lower Bound

We first prove that the trivial upper bound on $R_{\mathcal{S}}$ is asymptotically tight.

**Theorem 1.** *There exists an instance, such that for any distribution over stable matchings, one worker only receives a $2/N$ fraction of their optimal stable share, i.e., $R_{\mathcal{S}} \geq \frac{N}{2} = \Omega(N)$.*

To prove Theorem 1, we construct an instance with $N/2$ highly-skilled workers and $N/2$ regular workers, such that every stable matching can satisfy at most one regular worker at a time, proving that $R_{\mathcal{S}} \geq N/2$. The formal proof is deferred to Appendix B.

However, a closer look at our instance reveals that all regular workers can be satisfied in a single (non-stable) matching (See Remark 4 in Appendix B). Thus, we turn our attention to distribution over (possibly non-stable) matchings, and the ratio $R_{\mathcal{M}}$. Theorem 2 shows that if we extend the support of $D$ to include all matchings, i.e., $D \in \Delta(\mathcal{M})$, the ratio $R_{\mathcal{M}}$ is still lower bounded by $\log N$.

**Theorem 2.** *There exists an instance s.t. for any distribution over (possibly non-stable) matchings, one worker only receives a $1/\Omega(\log N)$ fraction of their optimal stable share, i.e., $R_{\mathcal{M}} = \Omega(\log N)$.*

To prove Theorem 2, we recursively construct instances with global ranking of jobs over workers, and each worker could be assigned to a job they like, but such that the number of workers grows logarithmically faster than the number of valuable jobs, proving that each worker can only receive a logarithmic fraction of their optimal stable share. The full proof could be found in Appendix B.

## 3.2 Upper Bound

We show that the logarithmic ratio obtained in Theorem 2 is asymptotically tight, even if we consider distributions over internally stable matchings.

**Theorem 3.** *For any problem instance, there exists a distribution $D$ over internally stable matchings s.t. all workers only receive a $1/\mathcal{O}(\log N)$ fraction of their optimal stable share, i.e., $R_{\mathcal{I}} = \mathcal{O}(\log N)$.*

We prove Theorem 3 by constructing an offline approximation oracle (Algorithm 1), which generates a uniform distribution over $m$ internally stable matchings $\tilde{\mu}_1, \ldots, \tilde{\mu}_m$. Each worker $w$ is matched in exactly one matching $\tilde{\mu}_i$, the key technical insight is that setting $m > \log_2 N + 1$ ensures $\boldsymbol{U}_D(w) = \boldsymbol{U}(w, \tilde{\mu}_i(w))/m \geq \boldsymbol{U}^*(w)/m$. To prove this, we construct a directed forest where nodes represent workers who prefer a stable matching over the algorithm's output, and edges capture conflicts where workers compete for the same job copies under different matchings. By exploiting the tree structure and stability constraint, the proof shows that if any worker were worse off, the graph would imply an exponential growth in the number of workers. For more details, please refer to Appendix C.

**Remark 1.** *The distribution computed by Algorithm 1 is not only "ex-post" internally stable, but also "ex-ante" (externally) stable, in the sense that no worker has justified envy towards any other worker's (randomized) allocation.*

**Remark 2.** *In Algorithm 1, each worker is assigned a job with a probability of $1/m$. Under such an allocation, some matchings in the support only assign a subset of jobs. In practice, if some job $a$ is not allocated in a matching $\tilde{\mu}_j$, but is allocated to worker $w$ in $\tilde{\mu}_i$, we can give $a$ to $w$ in $\tilde{\mu}_j$ without breaking internal stability of $\tilde{\mu}_j$. This post-processing is a Pareto improvement of our solution.*

---

**Algorithm 1** Internally Stable Matchings for Matching Market with Indifference

---

**Input:** $N$ workers, $K$ jobs, Utility matrix $\boldsymbol{U}$ that encodes the preference of workers over jobs, strict preference list $P_a$ of jobs over workers, a positive number $m$.

1: For each job $a \in \mathcal{A}$, duplicate it $m$ times and denote the $i$-th copy as $a^{(i)}$.
2: Each replica $a^{(i)}$ shares the same preference $P_a$ as the original job $a$.
3: For each worker $w$, define an ordering $P_w$, by sorting jobs $a_k^{(i)}$ by decreasing utility $\boldsymbol{U}(w, a)$, breaking ties in favour of lower duplication index $i$, then in favour of lower index $k$. That is,

$$a_k^{(i)} \succ_{P_w} a_\ell^{(j)} \quad \Leftrightarrow \quad \begin{cases} \boldsymbol{U}(w, a_k) > \boldsymbol{U}(w, a_\ell) & \text{or} \\ \boldsymbol{U}(w, a_k) = \boldsymbol{U}(w, a_\ell) \text{ and } i < j & \text{or} \\ \boldsymbol{U}(w, a_k) = \boldsymbol{U}(w, a_\ell) \text{ and } i = j \text{ and } k < \ell \end{cases}$$

4: Run Gale-Shapley algorithm on $P_w$ and $P_a$ to compute a worker-optimal stable matching $\tilde{\mu}$.
5: For each $i \in [m]$, build a matching $\tilde{\mu}_i$, which matches each job $a$ with $\tilde{\mu}_i(a) := \tilde{\mu}(a^{(i)})$.
**Output:** The distribution $D$ which selects each matching $\tilde{\mu}_i$ with probability $1/m$.

---

Finally, we show that Algorithm 1 cannot be manipulated by a worker who mis-reports her preferences to obtain a distribution that gives them a higher utility, whereas the proof is deferred to Appendix C.3.

**Theorem 4.** *Algorithm 1 is dominant strategy incentive compatible: for every utility matrices $\boldsymbol{U}$ and $\boldsymbol{U}'$ that differ only on the row of worker $w$, let $D$ and $D'$ be the distributions computed by Algorithm 1, then $\boldsymbol{U}_D(w) \geq \boldsymbol{U}_{D'}(w)$.*

## 4 Robustness and $\epsilon$-Stability

In Section 3, we present an asymptotically tight algorithm for approximating the optimal stable share in markets with ties under stability. However, exact stability often proves too rigid for real-world applications where preferences may fluctuate slightly. We therefore introduce $\epsilon$-stability, which tolerates blocking pairs with utility gains below a threshold $\epsilon$. This relaxation yields robust matching resilient to preference perturbations while maintaining theoretical guarantees.

**Definition 3** ($\epsilon$-Stability)**.** *Given $\epsilon \geq 0$, a matching $\mu$ is $\epsilon$-stable if there is no $\epsilon$-blocking pair $(w, a)$ such that $w \succ_a \mu(a)$ and $\boldsymbol{U}(w, a) > \boldsymbol{U}(w, \mu(w)) + \epsilon$.*

The notion of $\epsilon$-stability is a relaxation of weak stability, where setting $\epsilon = 0$ makes it equivalent to weak stability (Definition 1). In general, $\epsilon$-stable matching is not unique, and there is not a single $\epsilon$-stable matching that simultaneously maximizes the utilities for all workers. Therefore, similar to matching markets with ties, we define $\mathcal{S}_\epsilon := \{\mu : \mu \text{ is an } \epsilon\text{-stable matching}\}$, and we call $a$ a *valid $\epsilon$-stable match* of worker $w$ if there exists an $\epsilon$-stable matching matches $w$ with $a$, and it is the *optimal $\epsilon$-stable match* of worker $w$ if it is the most preferred valid $\epsilon$-stable match, i.e., there exists a matching $\mu_\epsilon^* \in \mathcal{S}_\epsilon$ such that $\mu_\epsilon^*(w) = a$ and $\boldsymbol{U}(w, \mu_\epsilon^*(w)) = \max_{\mu \in \mathcal{S}_\epsilon} \boldsymbol{U}(w, \mu(w))$. And we say $\boldsymbol{U}(w, \mu_\epsilon^*(w))$ is the *optimal $\epsilon$-stable share* for worker $w$, denoted as $\boldsymbol{U}_\epsilon^*(w)$.

Algorithm 2 (see Appendix D) generalizes Algorithm 1 with a different workers' preference profiles generation. It outputs a randomized matching that achieves an expected utility within a $\log N$ factor of the optimal $\epsilon$-stable share, plus an $\epsilon$-additive error.

**Theorem 5.** *Given any utility matrix $\boldsymbol{U}$, parameter $m = \lfloor \log_2 N + 2 \rfloor$, and the instability tolerance $\epsilon \geq 0$, Algorithm 2 computes a distribution $D \in \Delta(\mathcal{I})$, such that $\boldsymbol{U}_D(w) \geq \frac{\boldsymbol{U}_\epsilon^*(w)}{m} - \epsilon, \forall w \in \mathcal{W}$.*

The proof of Theorem 5 is deferred to Appendix D.2. Interestingly, the distribution $D$ randomizes over internally stable matchings, which do not depend on $\epsilon$.

In labor markets, worker preferences are typically estimated with uncertainty via i.i.d. observations, we construct utility uncertainty sets using concentration inequalities. Theorem 6 shows that for any utility matrix in such a set $\mathcal{U}$, Algorithm 2 produces a random matching guaranteeing each

worker a logarithmic approximation to their optimal share within $\mathcal{U}$, where the proof is deferred to Appendix D.4.

**Theorem 6.** *Given an uncertainty set $\mathcal{U}$, the optimal stable share within $\mathcal{U}$ is*

$$\mathcal{U}^*(w) := \sup_{\boldsymbol{U} \in \mathcal{U}} \max_{\mu \in \mathcal{S}^{\boldsymbol{U}}} \boldsymbol{U}(w, \mu(w)), \quad \forall w \in \mathcal{W}. \tag{2}$$

*We define the center $\hat{\boldsymbol{U}}$ of the set $\mathcal{U}$ as $\hat{\boldsymbol{U}}(w, a) = \frac{\inf_{\boldsymbol{U} \in \mathcal{U}} \boldsymbol{U}(w,a) + \sup_{\boldsymbol{U} \in \mathcal{U}} \boldsymbol{U}(w,a)}{2}$, and the uncertainty parameter as $\epsilon = 2 \cdot \sup_{\boldsymbol{U}_1, \boldsymbol{U}_2 \in \mathcal{U}} ||\boldsymbol{U}_1 - \boldsymbol{U}_2||_{\max}$. Algorithm 2 with input $\hat{\boldsymbol{U}}$, $m = \lfloor \log_2 N + 2 \rfloor$, and $\epsilon$ outputs a distribution $D \in \Delta(\mathcal{I})$ such that $\boldsymbol{U}_D(w) \geq \frac{\mathcal{U}^*(w)}{m} - \epsilon, \forall w \in \mathcal{W}$.*

Example 2 illustrates an application of Theorem 6 to batch learning problems.

**Example 2** (Batch learning). *Suppose that we have a dataset of size $T$, where each data point $\boldsymbol{U}$ is a noisy observation of the ground-truth utility matrix $\tilde{\boldsymbol{U}}$, i.e., each $\boldsymbol{U}(i, j)$ is sampled from a 1-sub-Gaussian distribution with mean $\tilde{\boldsymbol{U}}(i, j)$. Given a parameter $\delta$, set $\epsilon = 2\sqrt{\ln(\frac{1}{\delta})/T}$, and define the uncertainty set for each entry $(w, a)$ as $\mathcal{U}_{w,a} = \left\{ \boldsymbol{U}(w, a) : |\boldsymbol{U}(w, a) - \hat{\boldsymbol{U}}(w, a)| \leq \epsilon/2 \right\}$, and $\mathcal{U} = \bigotimes_{(w,a) \in \mathcal{W} \times \mathcal{A}} \mathcal{U}_{w,a}$, where $\hat{\boldsymbol{U}}$ is the empirical mean utility matrix computed from the dataset. The OSS within the uncertainty set $\mathcal{U}^*(w)$ could be defined as in Eq.(2). By Lemma 2, we know that with probability $1 - \delta$, the ground-truth utility matrix $\tilde{\boldsymbol{U}} \in \mathcal{U}$, and hence $\tilde{\boldsymbol{U}}^*(w) \leq \mathcal{U}^*(w)$. Therefore, by running Algorithm 2 with the empirical mean utility matrix as input, and set $\epsilon = 2\sqrt{\ln(\frac{1}{\delta})/T}$, $m = \lfloor \log_2 N + 2 \rfloor$, we have w.p. $1 - \delta$ that the corresponding output distribution $D$ over matchings satisfies $\boldsymbol{U}_D(w) \geq \frac{\tilde{\boldsymbol{U}}^*(w)}{\lfloor \log_2 N+2 \rfloor} - 2\sqrt{\ln(\frac{1}{\delta})/T}$ for all $w \in \mathcal{W}$.*

## 5  Bandit Learning in Matching Markets

Example 2 demonstrates the application of our offline oracle to learning problems. We now transition to an *online learning* setting, framing the matching market as a *multi-player bandit problem* to show how the offline results naturally connect learning scenarios both with and without statistical ties.

In online marketplaces, companies can evaluate workers through interviews, but typically lack prior knowledge of worker preferences over jobs. Still, by leveraging repeated matching opportunities, these preferences can be learned through ex-post evaluations. Recent work models this as a multi-armed bandit (MAB) problem [42, 43, 8, 37], where workers ("players") and jobs ("arms") interact over $T$ rounds. Each round, the company outputs a matching $\mu_t$ assigning jobs to workers and observes 1-subgaussian rewards $X_i(t)$ for matched pairs $(w_i, \mu_t(w_i))$ with mean $\boldsymbol{U}(w_i, \mu_t(w_i)) \in [0, 1]$. Following bandit matching literature, we assume $N \leq K$ (more jobs than workers) to ensure matching feasibility. If $N > K$, we can extend the problem by adding zero-utility jobs or randomly assigning unmatched workers.

The company seeks to learn the worker-optimal stable matching $\mu^*(w_i)$ through interactions. Specifically, it aims to minimize the worker-optimal stable regret for each $w_i \in \mathcal{W}$, defined as the cumulative reward difference between being matched with $\mu_i^*$ and that $w_i$ receives over $T$ rounds:

$$Reg_i(T) = T \cdot \boldsymbol{U}^*(w_i) - \mathbb{E}\left[ \sum_{t=1}^{T} X_i(t) \right]. \tag{3}$$

The expectation is taken over the randomness of the received reward and the allocation strategy.

Prior work on minimizing worker-optimal stable regret focuses exclusively on tie-free markets [42, 8, 37], rendering their results inapplicable when preferences contain ties. Crucially, existing regret bounds scale as $1/\Delta^2$, where $\Delta$ is the minimum utility gap across all workers $w$ and jobs $a$, i.e., $\Delta = \min_w \min_{a,a'} |\boldsymbol{U}(w, a) - \boldsymbol{U}(w, a')|^2$. As shown in Example 2 in [42], this dependence is fundamental – achieving sublinear regret requires $\Delta = \omega(1/\sqrt{T})$.

When the benchmark is unachievable (computationally or statistically), prior work adopts $\alpha$-approximation regret to ensure sublinear regret relative to an $\alpha$-fraction of the benchmark [31, 54, 14].

---

[2]While definitions of $\Delta$ vary slightly across works, this strongest version generalizes to other formulations.

In our setting, since ties prevent all workers from simultaneously achieving their optimal stable share, we assume access to an offline oracle that, given utility matrix $\boldsymbol{U}$, outputs a randomized matching guaranteeing each worker at least an $1/\alpha$ of $\boldsymbol{U}^*(w)$ in expectation, with additional error $\epsilon$. Formally,

**Definition 4** (($\boldsymbol{\alpha}$, $\epsilon$)-Approximation Oracle)**.** *An ($\boldsymbol{\alpha}$, $\epsilon$)-approximation oracle takes a rectangular uncertainty set $\mathcal{U}$ with width $\epsilon$ as input and returns a (randomized) matching $\tilde{\mu}$ satisfying: $\mathbb{E}\left[\boldsymbol{U}_{\tilde{\mu}}(w)\right] \geq \boldsymbol{\alpha}^{\mathcal{U}}(w) \cdot \mathcal{U}^*(w) - \epsilon$ for every worker $w$, where $\boldsymbol{\alpha}^{\mathcal{U}} \in (0,1]^N$ is a worker-specific approximation ratio vector (often simplified to $\boldsymbol{\alpha}$). If $\boldsymbol{\alpha}^{\mathcal{U}}(w) = \alpha$ is uniform across workers and independent of $\mathcal{U}$, we call it an $(\alpha, \epsilon)$-approximation oracle,*

For example, Algorithm 2 guarantees that for any input utility matrix $\boldsymbol{U}$, $\boldsymbol{\alpha}^{\boldsymbol{U}}(w) \geq 1/\lfloor \log_2 N + 2 \rfloor$. With ties, our regret metric should not compare against the OSS each time, but against an $\alpha$-fraction of the optimal stable share, since the offline oracle can only guarantee this fraction in expectation:

$$Reg_i^\alpha(T) = \alpha T \cdot \boldsymbol{U}^*(w_i) - \mathbb{E}\left[\sum_{t=1}^T X_i(t)\right], \tag{4}$$

where $\alpha \in (0,1]$ is the approximation ratio given by the offline oracle. When we want to emphasize that the observations $X(t)$ come from a distribution $\boldsymbol{\nu}$, we write $Reg_i(T;\boldsymbol{\nu})$ and $Reg_i^\alpha(T;\boldsymbol{\nu})$.

For markets without ties, [37] achieves a stable regret of $\mathcal{O}(K \ln T/\Delta^2)$, matching the $\Omega(N \ln T/\Delta^2)$ lower bound [52] in $T$ and $\Delta$. We seek a *best-of-both-worlds* guarantee, i.e., an algorithm that attains $Reg_i(T) = \mathcal{O}(\ln T/\Delta^2)$ when $\Delta = \omega(1/\sqrt{T})$, and $Reg_i^\alpha(T) = o(T)$ when $\Delta = \mathcal{O}(1/\sqrt{T})$.

## 5.1 Algorithm: Explore-then-Choose-Oracle

We present our algorithm, Explore-then-Choose-Oracle (ETCO, Algorithm 3 in Appendix E), and summarize it here. The algorithm consists of two phases. In each round of the *exploration phase*, the company allocates a job to every worker in a round-robin way to estimate their utilities accurately. In the second phase, the company checks for plausible ties in utilities. If none exists, it computes a matching using GS algorithm; otherwise, it uses the approximation oracle. In subsequent rounds, jobs are allocated based on the chosen oracle's output.

In the exploration phase, the company allocates jobs to workers in a round-robin way, according to the index of the workers. In this way, every $K$ rounds, each worker is matched to every job exactly once. The maximal number of exploration rounds is bounded by a parameter $T_0$. After each allocation, based on the observation, we update the estimated utility $\hat{\boldsymbol{U}}(i, \mu_t(i)) = \frac{\hat{\boldsymbol{U}}(i,\mu_t(i)) \cdot T_{i,\mu_t(i)} + X_{i,\mu_t(i)}(t)}{T_{i,\mu_t(i)}+1}$, and the observation count of worker $w_i$ and job $\mu_t(i)$ as $T_{i,\mu_t(i)} = T_{i,\mu_t(i)} + 1$. The company also builds a confidence set for each utility estimate, ensuring the true expected utility is included with high probability. Particularly, the confidence interval (CI) for worker $w_i$'s preference utility over job $a_j$ is $[LCB_{i,j}, UCB_{i,j}]$, with the upper and lower confidence bounds defined as

$$UCB_{i,j} = \hat{\boldsymbol{U}}(i,j) + \sqrt{\frac{6\ln T}{\max\{T_{i,j}, 1\}}}, \quad LCB_{i,j} = \hat{\boldsymbol{U}}(i,j) - \sqrt{\frac{6\ln T}{\max\{T_{i,j}, 1\}}}. \tag{5}$$

When confidence sets for jobs $a_j$ and $a_{j'}$ are disjoint ($LCB_{i,j} > UCB_{i,j'}$ or vice versa), we can determine worker $w_i$'s strict preference between them. If all top-$N$ job CIs for $w_i$ become disjoint, we recover the true preference with high probability. If this occurs for all workers before the exploration phase $T_0$ ends, we switch to the Gale-Shapley oracle for exploitation, as no top-$N$ ties exist w.h.p. Otherwise, remaining CI overlaps indicate potential ties, triggering our approximation oracle instead.

## 5.2 Theoretical Analysis

Before stating the regret guarantee for ETCO algorithm, we first give a formal definition of the minimum preference gap, which measures the hardness of the learning problem.

**Definition 5** (Minimum Preference Gap)**.** *For each worker $w_i$ and job $a_j \neq a_{j'}$, let $\Delta_{i,j,j'} = |\boldsymbol{U}(i,j) - \boldsymbol{U}(i,j')|$ be the preference gap for $w_i$ between $a_j$ and $a_{j'}$. Let $r_i$ be the preference ranking of worker $w_i$ and $r_{i,k}$ be the $k$-th preferred job in $w_i$'s ranking for $k \in [K]$. Define $\Delta_{\min} = \min_{i \in [N]; k \in [N]} \Delta_{i,r_{i,k},r_{i,k+1}}$ as the minimum preference gap among all workers and their first $(N+1)$-ranked jobs.*

Next, we present upper bounds for the worker-optimal stable regret for each worker when using ETCO.

**Theorem 7** (Upper Bound). *Following the ETCO algorithm with exploration phase of length $T_0$ and an $\left(\alpha, 2\sqrt{\frac{6K \ln T}{T_0}}\right)$-approximation oracle, for $w_i \in \mathcal{W}$, we have that*

$$Reg_i(T) = \mathcal{O}\left(\frac{K \ln T}{\Delta_{\min}^2}\right) \qquad if \quad \Delta_{\min} > \sqrt{\frac{96 K \ln T}{T_0}} = \Omega\left(\sqrt{\frac{K \ln T}{T_0}}\right), \quad (6)$$

$$Reg_i^\alpha(T) \leq 2\alpha T_0 + \mathcal{O}\left(T\sqrt{\frac{K \ln T}{T_0}}\right) \qquad if \quad \Delta_{\min} \leq \sqrt{\frac{96 K \ln T}{T_0}} = \mathcal{O}\left(\sqrt{\frac{K \ln T}{T_0}}\right). \quad (7)$$

See Appendix G for the proof. Our bound exhibits two regimes: (1) **large-$\Delta$ regime**: when $\Delta_{\min}$ is large, the exploration phase learns the top-$(N+1)$ job preferences w.h.p. before $T_0$, enabling exact worker-optimal stability via Gale-Shapley in exploitation. This reduces to ETGS [37] under centralization; (2) **small-$\Delta$ / tied regime**: for small $\Delta_{\min}$ or exact ties, worker-optimal stability is unattainable; instead, implementing an approximation oracle guarantees an $\alpha$-approximation regret sublinear in $T$.

When $\Delta_{\min}$ is sufficiently large, our upper bound matches the $\Omega(N \ln T/\Delta_{\min}^2)$ lower bound [52] for serial dictatorship markets (where all jobs share identical preferences). This tightness, however, comes at a fundamental trade-off: Theorem 8 shows that extending sublinear regret guarantees to wider ranges of $\Delta_{\min}$ unavoidably worsens approximation regret in small- or no-gap regimes.

Prior to presenting our trade-off lower bound, we formally define two key concepts. The *Pareto-optimal stable matching set* $\mathcal{S}_{opt}^U$, comprising stable matchings where no worker's utility can be strictly improved without harming another worker, and the *relevant preference utility gap* $\Delta_{rel}$, representing the maximum utility perturbation that preserves $\mathcal{S}_{opt}^U$.

**Definition 6** (Pareto-optimal Stable Matching Set). *Given a utility matrix $U$, the worker-optimal Pareto-optimal stable matching set $\mathcal{S}_{opt}^U$ is the set of all matchings $\mu$ such that: 1) $\mu$ is stable; 2) If there exists a stable matching $\mu'$ and a worker $w$ such that $U(w, \mu'(w)) > U(w, \mu(w))$, then for some $w' \neq w$, it holds that $U(w', \mu'(w')) < U(w', \mu(w'))$.*

**Definition 7** (Relevant Utility Gap). *Given a utility matrix $U$, the relevant preference gap $\Delta_{rel}$ is*

$$\Delta_{rel} := \inf\left\{\varepsilon : \exists\, i \in [N], j \in [K], \tilde{U}(i,j) \in [U(i,j) - \varepsilon, U(i,j) + \varepsilon] \text{ s.t. } \mathcal{S}_{opt}^U \neq \mathcal{S}_{opt}^{\tilde{U}}\right\}. \quad (8)$$

By definition, $\Delta_{rel} \geq 0$. When $\mathcal{S}_{opt}^U$ contains multiple matchings, $\Delta_{rel} = 0$ since any perturbation acts as a tie-breaker, eliminating at least one matching from $\mathcal{S}_{opt}^U$ (by the uniqueness of worker-optimal stable matching in tie-free markets). Furthermore, $\Delta_{rel} \geq \Delta_{\min}$ because perturbations smaller than $\Delta_{\min}$ cannot alter any worker's top-$N$ preferences or the worker-optimal matching[3].

**Theorem 8** (Trade-off between Regret and Approximation Regret). *Let $\delta \in (0, \frac{1}{2})$ and fix $N = K = 4$. Consider the class of instances with a large relevant utility gap, denoted as $\mathcal{E}_\ell(T)$, i.e., for any instance $\boldsymbol{\nu} \in \mathcal{E}_\ell(T)$, we have $\Delta_{rel}^{\boldsymbol{\nu}} \geq cT^{-1/2+\delta}$ for some absolute $c > 0$. Assume that an algorithm $\pi$ guarantees sublinear regret for all workers, for all $\boldsymbol{\nu} \in \mathcal{E}_\ell(T)$. Then there exists an instance such that this algorithm suffers $\Omega(T^{1-2\delta})$ approximation regret for some worker when $\Delta_{rel} = 0$ w.r.t the best approximation ratio $\boldsymbol{\alpha}^*$ for this instance, i.e.,*

$$If\, \forall\, w_i \in \mathcal{W}, \quad \limsup_{T \to \infty} \frac{\sup_{\boldsymbol{\nu} \in \mathcal{E}_\ell(T)} Reg_i(T; \boldsymbol{\nu})}{T} = 0,$$

$$\implies \exists\, w_i \in \mathcal{W}, \boldsymbol{\nu}' \quad s.t. \quad \Delta_{rel}^{\boldsymbol{\nu}'} = 0, \text{ and } Reg_i^{\boldsymbol{\alpha}^*(w_i)}(T; \boldsymbol{\nu}') = \Omega(T^{1-2\delta}), \quad (9)$$

*where $\boldsymbol{\alpha}^*(w_i) = \max\left\{\boldsymbol{\alpha}(w_i) : \boldsymbol{\alpha}(w) \geq 1/R_{\mathcal{M}}^U, \forall\, w \in \mathcal{W}\right\}$, for any $w_i \in \mathcal{W}$, and $R_{\mathcal{M}}^U$ is the OSS-ratio on matchings with a given utility matrix $U$.*

The proof appears in Appendix H. We construct two serial dictatorship instances with 4 workers and 4 jobs each, whose utility matrices differ in only one entry (for the highest-priority worker), yielding $\Delta_{rel} = 0$. The first instance evaluates $\alpha^*$-approximation regret, while the second analyzes standard stable regret. Crucially, this single entry difference *completely alters* the benchmark utilities for the other three workers. Thus, one of the two cases happens: (1) **under-sampling**: without

---

[3] Actually, if we assume an oracle that can determine whether there is a unique worker-optimal stable matching within the uncertainty set, we can prove a similar upper bound as in Theorem 7 with $\Delta_{\min}$ replaced by $\Delta_{rel}$.

enough samples of the differing entry, at least one worker incurs linear approximation regret; (2) **over-sampling**: After $T_0$ samples of the differing entry, at least one of the remaining workers suffers $\Omega(T_0)$ approximation regret.

Theorem 8 establishes an inherent trade-off between large-gap and small / no-gap regimes: as $\delta \to 0$, sublinear regret in the former necessitates linear approximation regret in the latter. Consequently, the exploration length $T_0$ of ETCO algorithm critically determines regime-specific performance. We provide two $T_0$ choices and their corresponding regret bounds.

**Corollary 1.** *Following ETCO algorithm with $T_0 = T^{2/3} (K \ln T)^{1/3}$, for $w_i \in \mathcal{W}$, we have*

$$Reg_i(T) = \mathcal{O}\left(\frac{K \ln T}{\Delta_{\min}^2}\right) \text{ if } \Delta_{\min} = \tilde{\Omega}\left(T^{-\frac{1}{3}}\right); Reg_i^{\alpha}(T) = \mathcal{O}\left((K \ln T)^{\frac{1}{3}} T^{\frac{2}{3}}\right) \text{ if } \Delta_{\min} = \tilde{\mathcal{O}}\left(T^{-\frac{1}{3}}\right).$$

Choosing $T_0 = T^{2/3} (K \ln T)^{1/3}$ yields the optimal approximation regret upper bound in the small gap regime when implementing explore-then-commit type algorithms. However, this choice is not satisfiable when $\Delta_{\min} \in [\tilde{\Omega}\left(T^{-1/2}\right), \tilde{\mathcal{O}}\left(T^{-1/3}\right)]$, since setting $T_0$ as such cannot guarantee detection of instances when $\Delta_{\min}$ falls in this intermediate regime. For these cases, we must resort to the approximation oracle during exploitation. Cruicially, since the oracle's solution differs by a constant factor from the Gale-Shapley optimal, each exploitation round incurs constant regret when measured against Eq.(3), resulting in an overall linear regret.

**Corollary 2.** *Following ETCO algorithm with $T_0 = \frac{T}{2 \ln T}$, for $w_i \in \mathcal{W}$, we have*

$$Reg_i(T) = \mathcal{O}\left(\frac{K \ln T}{\Delta_{\min}^2}\right) \text{ if } \Delta_{\min} = \tilde{\Omega}\left(T^{-\frac{1}{2}}\right); Reg_i^{\alpha - \frac{1}{\ln T}}(T) = \mathcal{O}\left(\sqrt{KT} \ln T\right) \text{ if } \Delta_{\min} = \tilde{\mathcal{O}}\left(T^{-\frac{1}{2}}\right).$$

Choosing $T_0 = T/(2 \ln T)$ yields the optimal regret that matches the lower bound for any $\Delta_{\min} = \tilde{\Omega}\left(T^{-1/2}\right)$. However, for $\Delta_{\min} = \tilde{\mathcal{O}}\left(T^{-1/2}\right)$, we can only guarantee sublinear approximation regret with an approximation ratio of $\alpha - 1/\ln T$ even when using an offline $\alpha$-approximation oracle.

**Remark 3.** *The approximation regret lower bound in Theorem 8 is both non-trivial and potentially of independent interest for bandit theory. While combinatorial bandits typically use approximation regret to circumvent computational limits (with statistical lower bounds focusing on 1-regret [15, 39, 48]), our result reveals a fundamental distinction: in matching markets, this approximation factor persists even given unlimited computational resources.*

## 6 Conclusion

In this paper, we study stable matching with one-sided indifference, modeled as a company assigning workers to jobs. Using a utility matrix to encode workers' potentially tied preferences over jobs, we define the *optimal stable share* (OSS) for each worker as the maximum utility achievable in any stable matching. To address fairness, we introduce the OSS-ratio, quantifying the fraction of the OSS a worker obtains under random matchings. We first analyze distributions over stable matchings, showing that a linear approximation to the OSS is trivial and asymptotically tight. For general matchings, we prove that no better than logarithmic approximation is possible. To achieve this bound, we propose a polynomial-time algorithm computing a distribution over *internally stable matchings*, which is asymptotically optimal in OSS ratio and dominant strategy incentive-compatible. Next, we extend our framework to settings where the utility matrix is uncertain but lies within a given uncertainty set. By incorporating $\epsilon$-stable matchings and relating them to perturbations of the utility matrix, we derive a logarithmic approximation with an additive $\epsilon$ error, matching the deterministic case. Finally, we explore online learning, where existing stable regret frameworks fail to handle tied preferences. Leveraging the OSS-ratio, we define $\alpha$-approximation stable regret and provide an algorithm whose upper bound matches the lower bound in the no-tied case. We further derive approximation regret bounds for small or no utility gaps and establish a fundamental trade-off between regret types, highlighting the need for careful exploration stopping time decisions.

Our work establishes the first instance-independent worker-optimal stable regret bound in bandit learning for matching markets, achieved through centralized job allocation. However, real-world marketplaces typically operate in decentralized settings where workers cannot directly coordinate. While the Gale-Shapley algorithm naturally decentralizes, extending our approximation guarantees to decentralized bandit learning remains an open challenge, which is an important direction for future research. Furthermore, exploring the application of our proposed algorithms to real-world datasets would be a valuable next step, as it would help address the practical challenge of stable matching when ties exist in preference rankings.

## Acknowledgements

This project has received funding from the European Union's Horizon 2020 research and innovation programme under the Marie Skłodowska-Curie grant agreement No 101034255. Shiyun Lin acknowledges the financial support from the China Scholarship Council (Grant No.202306010152). Vianney Perchet's research was supported in part by the French National Research Agency (ANR) in the framework of the PEPR IA FOUNDRY project (ANR-23-PEIA-0003) and through the grant DOOM ANR-23-CE23-0002. It was also funded by the European Union (ERC, Ocean, 101071601). Views and opinions expressed are however those of the author(s) only and do not necessarily reflect those of the European Union or the European Research Council Executive Agency. Neither the European Union nor the granting authority can be held responsible for them.

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

# A  Further Related Work

**Fractional Matchings**   Aside from *integral* matchings, *fractional* matchings have also attracted research interests due to their practical implications. For instance, in our running example, consider a *time-sharing* scenario [51], where each worker could spend five days a week at work. An integral matching requires every worker to work full-time on a single job, while fractional matchings allow them to switch among different jobs, making it natural in such situations. By the well-known Birkhoff-von-Neumann (BvN) theorem [10, 58], a fractional matching could be written as a convex combination of several integral matchings.

In the context of stable matching, fractional matching has also been studied. Considering purely ordinal preferences, several notions of stability have been proposed, such as *strong stability* [51], *ex-post stability* [51], and *fractional stability* [57]. In these works, the stable matching problem is formulated as a linear program, Teo and Sethuraman [55] showed that any fractional solution in the stable matching polytope is a convex combination of integral stable matchings. On the other hand, concerning purely cardinal preferences, Anshelevich et al. [4] proposes the notions of stability and $\varepsilon$-stability, while Caragiannis et al. [12] shows that the set of stable fractional matchings that satisfies the notion can be non-convex.

In this paper, we consider a two-sided market where the worker side has cardinal preferences while the job side has ordinal preferences. We do not concern the notions of fractional stable matchings, instead, we focus on finding a distribution over integral matchings such that it is fair in the sense that every worker could receive a certain fraction of its optimal stable share in expectation.

**Fair Division**   Fair division is the problem of dividing a set of items among several people in a fair manner. Steinhaus [53] pioneers this line of research and defines a share-based notion, i.e., *proportionality*, where each player gets a $1/N$ fraction of all items. Foley [20] and Varian [56] define *envy-freeness*, where no player prefers the bundle allocated to another player, and this notion is later generalized by Weller [59]. In two-sided matching markets, a stable matching eliminates *justified envy* [1]. Regarding the problem of sharing indivisible goods, share-based guarantees such as MMS [11] and envy-based guarantees such as EF1 [41, 11] or EFX [13] are proposed. Recent works have studied best-of-both-world fairness [6, 7, 19], providing random allocation with fairness guarantees both in expectation and for every realization.

**Approximation Regret in Bandit Learning**   In combinatorial bandit problems, *approximation regret* is often considered instead of the standard regret [31, 54, 22, 14, 47, 49]. The reason mainly lies in the complex reward structure and the computational intractability of the problem, i.e., rewards are often dependent on the combination of actions, leading to an exponentially large action space, which makes it computationally prohibitive to find the exact solution.

Besides computational intractability, there is another more fundamental reason for using the approximation regret framework in this paper. The stable regret is defined for each worker, which means the company aims to solve a multi-objective optimization problem while it couldn't satisfy everyone simultaneously. Consequently, the approximation regret serves as a compromise between fairness and efficiency.

# B  Lower bounds on OSS-ratio

## B.1  Lower Bound for Distributions over Weakly Stable Matchings

The proof idea of Theorem 1 could be illustrated through Figure 1.

*Proof of Theorem 1.* Assume that $N$ is even, let $\mathcal{W} = \{w_1, w_2, \cdots, w_N\}$ and $\mathcal{A} = \{a_1, a_2, \cdots, a_K\}$ with $K = \frac{N}{2} + 1$ and $w_1 \succ w_2 \succ \cdots \succ w_N$ for all the jobs. The utility matrix that encodes the preference of workers over jobs is as follows:

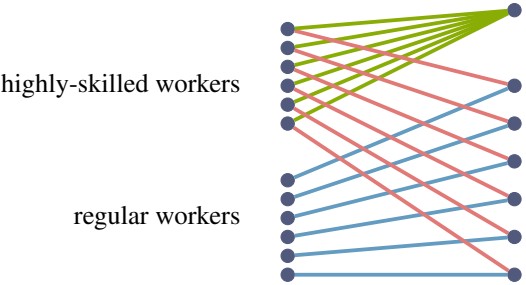

Figure 1: Lower bound on $R_{\mathcal{S}}$. All jobs have the same ordering over workers, from top to bottom. Any stable matching can be obtained by letting the first worker pick a job, then the second, etc. Hence, each stable matching contains at most one blue edge.

$$
\boldsymbol{U} = \left.\left(\begin{array}{ccccc}
1 & 0 & \cdots & 0 & 1 \\
0 & 1 & \cdots & 0 & 1 \\
\vdots & \vdots & \ddots & \vdots & \vdots \\
0 & 0 & \cdots & 1 & 1 \\
1 & 0 & \cdots & 0 & 0 \\
0 & 1 & \cdots & 0 & 0 \\
\vdots & \vdots & \ddots & \vdots & \vdots \\
0 & 0 & \cdots & 1 & 0
\end{array}\right)\begin{array}{c} \left.\begin{array}{c} \\ \\ \\ \\ \end{array}\right\} \frac{N}{2} \\ \left.\begin{array}{c} \\ \\ \\ \\ \end{array}\right\} \frac{N}{2} \end{array}\right. .
$$

In any stable matching, every worker $w_i$ in $\{w_1, w_2, \cdots, w_{N/2}\}$ must be assigned to $a_i$ or $a_{\frac{N}{2}+1}$, leading to a utility of 1 for them. Without loss of generality, let $\mu_i$ be the matching such that $\mu(w_i) = a_{\frac{N}{2}+1}$. Then in $\mu_i$, only worker $w_{\frac{N}{2}+i}$ would receive a utility of 1, by matching it to job $a_i$, while all workers in $\{w_{N/2+1}, \cdots, w_{N/2+i-1}, w_{N/2+i+1}, \cdots, w_N\}$ would be unmatched and receive a utility of 0. Indeed, for any $j \neq i$, the unique optimal match of worker $w_{N/2+j}$ is already taken by worker $w_j$.

For every stable matching, at most one of the workers in $\{w_{N/2+1}, w_{N/2+2}, \cdots, w_N\}$ could be assigned to their optimal match. Since there are $N/2$ such workers, then for any distribution $D$, there must be at least one of the workers for which the probability to be optimally matched is smaller than $2/N$, for this worker, it holds that $\frac{U^*(w)}{U_D(w)} \geq \frac{N}{2}$, which implies $R_{\mathcal{S}} \geq N/2$.

$\square$

**Remark 4.** *Theorem 1 shows that if we only consider random allocations of stable matchings, then in the worst case, workers could only expect $\mathcal{O}(1/N)$ profit share compared to their benchmark. On the other hand, define*

$$
\mu_1 = \{(w_i, a_i) : i \in \{1, 2, \cdots, N/2\}\},
$$
$$
\mu_2 = \{(w_{i+N/2}, a_i) : i \in \{1, 2, \cdots, N/2\}\}.
$$

*Here, $\mu_1$ is a stable matching while $\mu_2$ is non-stable. We construct a distribution $D$ as follows:*

$$
\mathbb{P}(D = \mu_1) = \frac{1}{2}, \quad \mathbb{P}(D = \mu_2) = \frac{1}{2},
$$

*then all workers have $\frac{U^*(w)}{U_D(w)} = 2$. This result implies that if we consider possibly non-stable matchings for the support of $D$, there is space for improvement on the OSS-ratio.*

## B.2 Lower Bound for Distributions over Matchings

The proof idea of Theorem 2 could be illustrated through Figure 2.

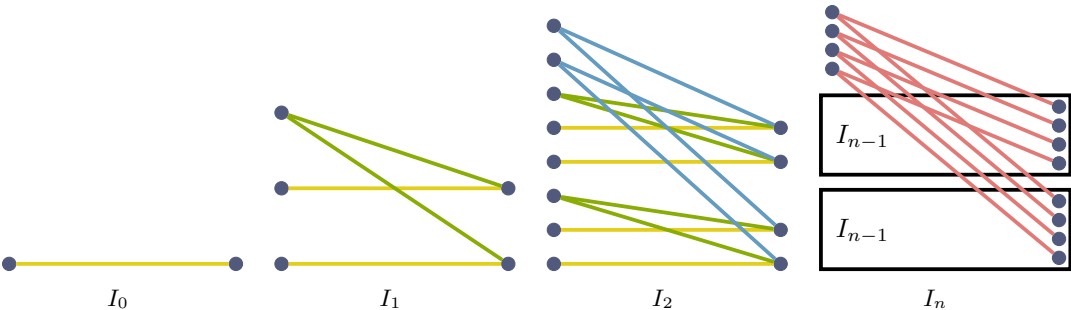

Figure 2: Lower bound on $R_{\mathcal{M}}$. In each example, left nodes represent workers while right nodes represent jobs. If there is an edge connecting a left node $w$ and a right node $a$, we have $\boldsymbol{U}(w, a) = 1$, and $\boldsymbol{U}(w, a) = 0$ otherwise. All the right nodes without edges connecting to them are hidden from the graph.

*Proof of Theorem 2.* Consider the following sequence of problem instances, as depicted in Figure 2. In each bipartite graph, left nodes represent the set of workers while the right nodes represent the set of jobs. Jobs share a global preference over the workers, with the topmost node being the most preferred and the preference decreasing from top to bottom. From the worker side, the utility matrix is binary. Specifically, for a worker-job pair $(w, a)$, $\boldsymbol{U}(w, a) = 1$ if $(w, a)$ is connected, while $\boldsymbol{U}(w, a) = 0$ otherwise. For example, $I_1$ is the graph representation of the problem instance of Example 1.

Instance $I_n$ is constructed recursively. Given $I_{n-1}$, we first duplicate this instance, denoted the replications as *upper class* and *lower class*, respectively. Take $K_{n-1}$ as the number of right nodes and $N_{n-1}$ as the number of left nodes in $I_{n-1}$, and denote these nodes as $\left\{a_1^u, a_2^u, \cdots, a_{K_{n-1}}^u\right\}$, $\left\{w_1^u, w_2^u, \cdots w_{N_{n-1}}^u\right\}$ and $\left\{a_1^\ell, a_2^\ell, \cdots, a_{K_{n-1}}^\ell\right\}$, $\left\{w_1^\ell, w_2^\ell, \cdots w_{N_{n-1}}^\ell\right\}$ for the upper and lower classes, respectively. Then, we introduce $K_{n-1}$ *prioritized workers* in $I_n$, who are uniformly more preferred by the jobs than the workers in the upper and lower classes. In particular, denote the set of prioritized workers as $\left\{w_1, w_2, \cdots, w_{K_{n-1}}\right\}$, we have

$$w_1 \succ w_2 \succ \cdots \succ w_{K_{n-1}} \succ w_1^u \succ w_2^u \succ \cdots \succ w_{N_{n-1}}^u \succ w_1^\ell \succ w_2^\ell \succ \cdots \succ w_{N_{n-1}}^\ell.$$

And for each $w_i$, we have $\boldsymbol{U}(w_i, a_i^u) = \boldsymbol{U}(w_i, a_i^\ell) = 1$, and $\boldsymbol{U}(w_i, a) = 0$ otherwise. We first prove by induction that the optimal-stable-share is $\boldsymbol{U}^*(w) = 1$ for any worker $w$. For $I_0$, the unique matching is stable. Suppose that for any worker $w$ in $I_{n-1}$, there exists at least one stable matching $\mu$ such that $\boldsymbol{U}(w, \mu(w)) = 1$. Then in $I_n$, all the prioritized workers could be matched in any stable matching. Furthermore, as long as they simultaneously choose to be matched to the jobs in the same class, the other class is free, and hence by induction assumption, every worker in that class gets a chance to be matched in at least one stable matching. Therefore, by breaking ties for the upper (lower) class, all workers in the lower (upper) class can be matched.

On the other hand, given that $\boldsymbol{U}^*(w) = 1$ for any $w$, the ratio $R_{\mathcal{M}}$ is equal to $\min_D \max_w \frac{1}{\boldsymbol{U}_D(w)}$ for these instances, Then, proving a lower bound on this quantity is equivalent to establishing an upper bound on $\max_D \min_w \boldsymbol{U}_D(w)$. In particular, we have $\max_D \min_w \boldsymbol{U}_D(w) \leq \max_D \frac{\sum_w \boldsymbol{U}_D(w)}{N}$, where $N$ is the number of left nodes. Now notice that $\sum_w \boldsymbol{U}_D(w)$ is the expected size of the matching under distribution $D$, since the utility is 1 when a worker is matched and 0 otherwise. We have $\sum_w \boldsymbol{U}_D(w) \leq K$ from the fact that the size of any matching is bounded by $K$. Combining the above derivation, we have

$$R_{\mathcal{M}} \geq \frac{N}{K}.$$

Finally, in instance $I_n$, by the recursive construction, we have

$$K_n = 2 \cdot K_{n-1},$$
$$N_n = 2 \cdot N_{n-1} + K_{n-1}.$$

Solving the recursive equation with the initial condition $K_0 = N_0 = 1$, we know that there are $2^n$ right nodes and $N = (n+2)2^{n-1}$ left nodes in $I_n$, which implies that $R_{\mathcal{M}} \geq n/2 + 1$. We rewrite $N = (n+2)2^{n-1}$ to obtain $2^n = 2N/(n+2)$ and we deduce that

$$N/n \leq 2^n \leq 2N.$$

Taking a logarithm in the inequalities, we have

$$\log_2 N - \log_2 n \leq n \leq 1 + \log_2 N.$$

And thus

$$n \geq \log_2 N - \log_2 n \geq \log_2 N - \log_2(1 + \log_2 N).$$

Therefore, $n = \Omega(\log N)$ and hence $R_{\mathcal{M}}$ is $\Omega(\log N)$. $\qquad\square$

## C  Upper Bound on OSS-ratio

### C.1  Procedure Illustration of Algorithm 1

We use Example 3 to illustrate the procedure stated in Algorithm 1.

**Example 3.** *Let $\mathcal{W} = \{w_1, w_2, w_3\}$, $\mathcal{A} = \{a_1, a_2, a_3\}$. We consider the following preference list $P_a$ of jobs over workers, and utility matrix $\boldsymbol{U}$ that encodes the preference of workers over jobs:*

$$
\begin{array}{ll}
a_1 & : w_2 \succ w_1 \succ w_3, \\
a_2 & : w_1 \succ w_3 \succ w_2, \\
a_3 & : w_1 \succ w_2 \succ w_3.
\end{array}
\qquad
\boldsymbol{U} = \begin{bmatrix} 1 & 1 & 0 \\ 0.5 & 0.1 & 0.1 \\ 0 & 0.8 & 0 \end{bmatrix}.
$$

*If $m = 2$, the preference profile $P_w$ generated from the algorithm is*

$$w_1 : a_1^{(1)} \succ a_2^{(1)} \succ a_1^{(2)} \succ a_2^{(2)} \succ a_3^{(1)} \succ a_3^{(2)},$$
$$w_2 : a_1^{(1)} \succ a_1^{(2)} \succ a_2^{(1)} \succ a_3^{(1)} \succ a_2^{(2)} \succ a_3^{(2)},$$
$$w_3 : a_2^{(1)} \succ a_2^{(2)} \succ a_1^{(1)} \succ a_3^{(1)} \succ a_1^{(2)} \succ a_3^{(2)}.$$

*Running Gale-Shapley algorithm on $P_w$ and $P_a$, the worker-optimal stable matching would be $\tilde{\mu} = \{(w_1, a_2^{(1)}), (w_2, a_1^{(1)}), (w_3, a_2^{(2)})\}$, and we can recover two internally stable matchings from $\tilde{\mu}$, i.e., $\tilde{\mu}_1 = \{(w_1, a_2), (w_2, a_1)\}$ and $\tilde{\mu}_2 = \{(w_3, a_2)\}$.*

### C.2  Proof of Theorem 3

In Algorithm 1, each worker $w$ is matched in exactly one matching $\tilde{\mu}_i$, where we call $i$ the index of $w$, denoted $\text{index}(w)$. In other words, the *index* of a worker is the index of the job she receives, that is, $\text{index}(w) = i$ if worker $w$ receives $a_j^{(i)}$ for some $j$.

**Definition 8.** *Given a problem instance $(\boldsymbol{U}, P_a)$, run Algorithm 1 with duplication number $m$ to generate the output distribution $D$. Then, for any stable matching $\mu$ with respect to $(\boldsymbol{U}, P_a)$, we define a graph $G_\mu = (V_\mu, E_\mu)$ where*

$$V_\mu := \{w \in \mathcal{W} \ : \ \boldsymbol{U}(w, \mu(w)) \geq m \cdot \boldsymbol{U}_D(w)\},$$
$$E_\mu := \{(w, w') \in V_\mu^2 \ : \ \mu(w) = \tilde{\mu}_j(w') \text{ where } j = \text{index}(w') < \text{index}(w)\}.$$

Informally, $G_\mu$ is the graph of workers who (weakly) prefer $\mu$ to their match in distribution $D$, where an edge $(w, w')$ means that $w'$ received a job that $w$ would have liked. Next, we show properties on the graph $G_\mu$, which we illustrate in Figure 3.

**Proposition 1.** *For any stable matching $\mu$, the following holds*

- *$G_\mu$ is a directed forest (there is no cycle and each vertex has at most one incoming edge),*

- *For every worker $w \in V_\mu$ with $i = \text{index}(w)$, and for every $1 \leq j < i$, there is a worker $w' \in V_\mu$ with $j = \text{index}(w')$ such that $(w, w') \in E_\mu$.*

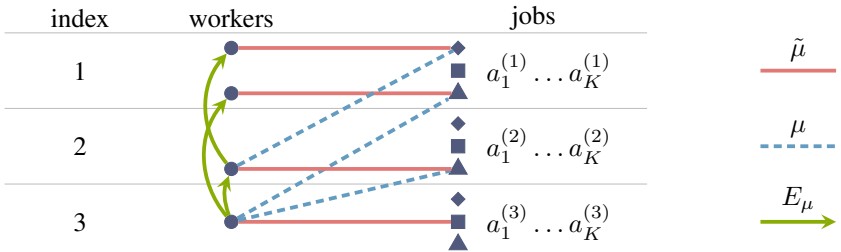

Figure 3: The graph $G_\mu = (V_\mu, E_\mu)$ is a directed forest. The matching $\tilde{\mu}$, computed in Algorithm 1 matches each worker to a single (copy of) job in $\tilde{\mu}$. In the stable matching $\mu$, each worker $w$ is connected to all copies of $\mu(w)$ which have lower index.

*Proof.* The graph $G_\mu$ is a directed forest by construction. Indeed, it has no cycle because edges connect workers to lower index workers. And every node has at most one incoming edge because $\tilde{\mu}$ matches each worker to at most one job, and $\mu$ matches each job to at most one worker.

Fix a worker $w \in V_\mu$ with $i = \text{index}(w)$, let $1 \leq j < i$, and let $a = \mu(w)$. By definition of $V_\mu$, worker $w$ weakly prefers $\mu$ to $D$, that is $\boldsymbol{U}(w, a) \geq m \cdot \boldsymbol{U}_D(w) = \boldsymbol{U}(w, \tilde{\mu}_i(w))$. By definition of $w$'s preference list $P_w$ in Algorithm 1, the lexicographic ordering gives that

$$a^{(j)} \succ_{P_w} \tilde{\mu}(w).$$

Because $\tilde{\mu}$ is a stable matching, it should not be blocked by the pair $(w, a^{(j)})$. Thus, there exists a worker $w' \in \mathcal{W}$ such that $\tilde{\mu}(w') = a^{(j)}$ and

$$w' \succ_a w.$$

Finally, because $\mu$ is a stable matching, it should not be blocked by the pair $(w', a)$, thus

$$\boldsymbol{U}(w', \mu(w')) \geq \boldsymbol{U}(w', a) = m \cdot \boldsymbol{U}_D(w'),$$

proving that $w' \in V_\mu$. Hence, there is an edge $(w, w') \in E_\mu$, which concludes the proof. $\square$

**Proposition 2.** *In the graph $G_\mu$, each node of index $i \geq 1$ can reach $2^{i-1}$ nodes (including itself).*

*Proof.* We show that the property holds by induction on $i$. The property trivially holds for $i = 1$. Let $i > 1$ such that there is a worker $w \in V_\mu$ with $\text{index}(w) = i$. Using Proposition 1, there is an edge $(w, w_j) \in E_\mu$ with $\text{index}(w_j) = j$ for every $1 \leq j < i$. Because the graph is a directed forest, the set of nodes reachable from each $w_j$ are disjoint. Thus, the number of nodes reachable from $w$ (including itself) is $1 + \sum_{j=1}^{i-1} 2^{j-1} = 2^{i-1}$. $\square$

Finally, we conclude with the proof that Algorithm 1 computes a distribution over internally stable matching which guarantees each worker a logarithmic fraction of their optimal stable share.

*Proof of Theorem 3.* Algorithm 1 first computes a stable matching $\tilde{\mu}$ for the instance with duplicated jobs, then build $m$ matchings $\tilde{\mu}_1, \ldots, \tilde{\mu}_m$. If there were a pair $(w, a)$ with $\text{index}(w) = i$ which blocks matching $\tilde{\mu}_i$, that is $\boldsymbol{U}(w, a) > \boldsymbol{U}(w, \tilde{\mu}_i(w))$ and $w \succ_a \tilde{\mu}_i(a)$, then $(w, a^{(i)})$ would block $\tilde{\mu}$, which is a contradiction. Thus, each matching $\tilde{\mu}_i$ is internally stable.

Now, let us assume, that there is a stable matching $\mu$ in which a worker $w$ with $\text{index}(w) = i$ receives $a = \mu(w)$ having utility $\boldsymbol{U}(w, a) > \boldsymbol{U}(w, \tilde{\mu}_i(w))$. In the matching $\tilde{\mu}$, job $a^{(m)}$ must be matched to some worker $w'$ such that $w' \succ_a w$, otherwise $(w, a)$ would block $\tilde{\mu}$. Moreover, we must have $\boldsymbol{U}(w', \mu(w')) \geq m \cdot \boldsymbol{U}_D(w')$ otherwise $(w', a)$ would be blocking $\mu$. Thus, there is a node $w' \in V_\mu$ of index $m$, which proves that there exists at least $2^{m-1}$ nodes, and thus that $N \geq 2^{m-1}$. By contrapositive, if we set $m > 1 + \log_2 N$, then we have $m \cdot \boldsymbol{U}_D(w) \geq \boldsymbol{U}^*(w)$ for every worker $w$, which concludes the proof.[4] $\square$

---

[4]Interestingly, we can show that $m > \log_2 N$ suffices because each $\mu(w)^{(i)}$ is matched in $\tilde{\mu}$ to a different worker $w_i \in V_\mu$, who can reach $2^{i-1}$ distinct workers in $G_\mu$, none of them being $w$ (as this would contradict $\tilde{\mu}$ being worker-optimal), which gives at least $2^m$ workers in total. However, for the sake of simplicity, we do not present this improved bound.

## C.3 Dominant Strategy Incentive Compatibility of Algorithm 1

*Proof of Theorem 4.* We will use the fact that when workers have strict preferences, Gale and Shapley's worker-proposing deferred acceptance procedure is dominant strategy incentive compatible, i.e., it is always optimal for workers to report their true preferences [17].

First, notice that for each worker $w$, the ranking $P_w$ used in Algorithm 1 aligns with her utility, ensuring that all copies of a higher-utility job are ranked above copies of lower-utility ones.

To see that it is optimal for a worker $w$ to report her true vector of utility, we will give her more strategic power, and we will let her choose her ranking $P'_w$ over all the duplicated jobs. By the incentive compatibility property of the deferred acceptance procedure with strict preferences, she cannot obtain any job ranked above $\tilde{\mu}(w)$ in $P_w$. And because $P_w$ is consistent with $w$'s utility, it is optimal to report $P'_w = P_w$. □

# D ε-Oracle for Approximated Worker Optimal Stable Matching

## D.1 ε-Oracle

---

**Algorithm 2** $\epsilon$-Oracle for Approximated Worker Optimal Stable Matching

---

**Input:** $N$ workers, $K$ jobs, Utility matrix $\boldsymbol{U}$ that encodes the preference of workers over jobs, strict preference profile $P_a$ of jobs, an integer $m \geq 1$, and the instability tolerance $\epsilon \geq 0$.

1: For each job $a \in \mathcal{A}$, duplicate it $m$ times and denote the $i$-th copy as $a^{(i)}$.
2: Each replica $a^{(i)}$ shares the same preference $P_a$ as the original job $a$.
3: For every worker $w$ and job $a^{(i)}$, define the utility

$$\boldsymbol{U}(w, a^{(i)}) := \boldsymbol{U}(w, a) - (i - 1)\epsilon$$

and use it to generate the workers' preference profile $P_w$ (breaking ties in favor of lower indices).
4: Run Gale-Shapley algorithm on $P_w$ and $P_a$ to compute a worker-optimal stable matching $\tilde{\mu}$.
5: For each $i \in [m]$, build a matching $\tilde{\mu}_i$, which matches each job $a$ with $\tilde{\mu}_i(a) := \tilde{\mu}(a^{(i)})$.
**Output:** The distribution $D$ which selects each matching $\tilde{\mu}_i$ with probability $1/m$.

---

## D.2 Proof of Theorem 5

Similarly to the proof of Theorem 3, we run Algorithm 2 and define the *index* of a worker as the index of the job she receives in $\tilde{\mu}$, that is, $\text{index}(w) = i$ if worker $w$ receives $a_j^{(i)}$ for some $j$.

**Definition 9.** *Given a problem instance $(\boldsymbol{U}, P_a)$, run Algorithm 2 with duplication number $m$ and instability tolerance $\epsilon$ to generate the output distribution $D$. Then, for any $\epsilon$-stable matching $\mu$ with respect to $(\boldsymbol{U}, P_a)$, we define a graph $G_\mu = (V_\mu, E_\mu)$ where*

$$V_\mu := \{w \in \mathcal{W} \mid \boldsymbol{U}(w, \mu(w)) \geq m \cdot \boldsymbol{U}_D(w) - \epsilon\},$$
$$E_\mu := \{(w, w') \in V_\mu^2 \mid \mu(w) = \tilde{\mu}_j(w') \text{ where } j = \text{index}(w') < \text{index}(w)\}.$$

Once again, $G_\mu$ is the graph of workers who prefer $\mu$ to their match in distribution $D$, where an edge $(w, w')$ means that $w'$ received a job that $w$ would have liked. Next, we show properties on the graph $G_\mu$.

**Proposition 3.** *For any stable matching $\mu$, we have that*

- *$G_\mu$ is a directed forest (there is no cycle and each vertex has at most one incoming edge),*

- *For every worker $w \in V_\mu$ with $i = \text{index}(w)$, and for every $1 \leq j < i$, there a worker $w' \in V_\mu$ with $j = \text{index}(w')$ such that $(w, w') \in E_\mu$.*

*Proof.* The proof is almost identical to that of Proposition 1. Fix a worker $w \in V_\mu$ with $i = \text{index}(w)$, let $1 \leq j < i$, and let $a = \mu(w)$. By definition of $V_\mu$, worker $w$ prefers $\mu$ to $D$, that is $\boldsymbol{U}(w, a) \geq$

$m \cdot \boldsymbol{U}_D(w) - \epsilon = \boldsymbol{U}(w, \tilde{\mu}_i(w)) - \epsilon$. By definition of $w$'s preference list $P_w$ in Algorithm 2, the lexicographic ordering gives that

$$a^{(j)} \succ_{P_w} \tilde{\mu}(w).$$

Because $\tilde{\mu}$ is a $\epsilon$-stable matching, it should not be blocked by the pair $(w, a^{(j)})$. Thus, there exists a worker $w' \in \mathcal{W}$ such that $\tilde{\mu}(w') = a^{(j)}$ and

$$w' \succ_a w.$$

Finally, because $\mu$ is a stable matching, it should not be blocked by the pair $(w', a)$, thus

$$\boldsymbol{U}(w', \mu(w)) \geq \boldsymbol{U}(w', a) - \epsilon = m \cdot \boldsymbol{U}_D(w') - \epsilon,$$

proving that $w' \in V_\mu$. Hence, there is an edge $(w, w') \in E_\mu$, which concludes the proof. □

We will once again use Proposition 2 to give a lower on the number of nodes in the graph $G_\mu$. Finally, we conclude with the proof that Algorithm 2 computes a distribution over internally $\epsilon$-stable matching which guarantees each worker a logarithmic fraction of their optimal stable share.

*Proof of Theorem 5.* Algorithm 2 first computes a stable matching $\tilde{\mu}$ for the instance with duplicated jobs, then build $m$ matchings $\tilde{\mu}_1, \ldots, \tilde{\mu}_m$. If there were a pair $(w, a)$ with $\mathrm{index}(w) = i$ which $\epsilon$-blocks matching $\tilde{\mu}_i$, that is $\boldsymbol{U}(w, a) > \boldsymbol{U}(w, \tilde{\mu}_i(w)) + \epsilon$ and $w \succ_a \tilde{\mu}_i(a)$, then $(w, a^{(i)})$ would block $\tilde{\mu}$, which is a contradiction. Thus, each matching $\tilde{\mu}_i$ is internally stable.

Now, let us assume, that there is a stable matching $\mu$ in which a worker $w$ with $\mathrm{index}(w) = i$ receives $a = \mu(w)$ having utility $\boldsymbol{U}(w, a) > \boldsymbol{U}(w, \tilde{\mu}_i(w)) + m\epsilon$. In the matching $\tilde{\mu}$, job $a^{(m)}$ must be matched to some worker $w'$ such that $w' \succ_a w$, otherwise $(w, a)$ would block $\tilde{\mu}$. Moreover, we must have $\boldsymbol{U}(w', \mu(w')) \geq m \cdot \boldsymbol{U}_D(w') - \epsilon$ otherwise $(w', a)$ would be blocking $\mu$. Using Proposition 2, there is a node $w' \in V_\mu$ of index $m$, which proves that there exists at least $2^{m-1}$ nodes, and thus that $N \geq 2^{m-1}$. By contrapositive, if we set $m > 1 + \log_2 N$, then we have $\boldsymbol{U}_D(w) \geq \boldsymbol{U}^*(w)/m - \epsilon$ for every worker $w$, which concludes the proof. □

### D.3 Robustness of $\epsilon$-stable matching

**Lemma 1.** *Fix the preferences of jobs over workers. Given two utility matrices $\boldsymbol{U}_1$ and $\boldsymbol{U}_2$ such that[5] $\|\boldsymbol{U}_1 - \boldsymbol{U}_2\|_{\max} < \frac{\epsilon}{2}$, then any stable matching $\mu$ for $\boldsymbol{U}_1$, is also $\epsilon$-stable with respect to $\boldsymbol{U}_2$.*

*Proof.* If a matching $\mu$ is stable with respect to $\boldsymbol{U}_1$, then for any $(w, a)$ pair such that $w \succ_a \mu(a)$, we must have

$$\boldsymbol{U}_1(w, a) \leq \boldsymbol{U}_1(w, \mu(w)), \tag{10}$$

from the definition of stable matching (Definition 1).

Since $\|\boldsymbol{U}_1 - \boldsymbol{U}_2\|_{\max} \leq \frac{\epsilon}{2}$, we have that for any $(w, a)$ pair, $|\boldsymbol{U}_1(w, a) - \boldsymbol{U}_2(w, a)| \leq \frac{\epsilon}{2}$. Therefore,

$$\boldsymbol{U}_2(w, a) \leq \boldsymbol{U}_1(w, a) + \frac{\epsilon}{2} \leq \boldsymbol{U}_1(w, \mu(w)) + \frac{\epsilon}{2} \leq \boldsymbol{U}_2(w, \mu(w)) + \epsilon, \tag{11}$$

where the first and the last inequality come from $\|\boldsymbol{U}_1 - \boldsymbol{U}_2\|_{\max} \leq \frac{\epsilon}{2}$, while the second inequality holds according to Eq.(10). Therefore, combining $w \succ_a \mu(a)$ and Eq.(11), we can conclude that matching $\mu$ is $\epsilon$-stable with respect to $\boldsymbol{U}_2$. □

### D.4 Proof of Theorem 6

For convenience, we denote $\mathcal{S}^{\boldsymbol{U}}$ (resp. $\mathcal{S}_\epsilon^{\boldsymbol{U}}$) the stable matchings ($\epsilon$-stable matchings) with respect to $\boldsymbol{U}$.

*Proof.* By running Algorithm 2 with $\boldsymbol{U} = \hat{\boldsymbol{U}}, \epsilon = \epsilon, m = \lfloor \log_2 N + 2 \rfloor$, from Theorem 5, we get

$$\boldsymbol{U}_D(w) \geq \frac{\hat{\boldsymbol{U}}_\epsilon^*(w)}{m} - \epsilon, \quad \forall w \in \mathcal{W}. \tag{12}$$

---

[5]We recall that the max norm of a matrix $A = (A_{i,j})$, is defined by $\|A\|_{\max} = \max_{i,j} |A_{i,j}|$.

By construction, any utility matrix $U \in \mathcal{U}$ satisfies $\|U - \hat{U}\|_{\max} \leq \frac{\epsilon}{2}$. From Lemma 1, we know that for any matching $\mu \in \mathcal{S}^U$, $\forall U \in \mathcal{U}$, we have $\mu \in \mathcal{S}_\epsilon^{\hat{U}}$, that is $\bigcup_{U \in \mathcal{U}} \mathcal{S}^U \subseteq \mathcal{S}_\epsilon^{\hat{U}}$. Therefore, for the optimal stable share, we have

$$\mathcal{U}^*(w) \leq \hat{U}_\epsilon^*(w), \quad \forall w \in \mathcal{W}. \tag{13}$$

Combining Eq.(12) and (13), the conclusion holds. $\qquad\square$

## E  Explore-then-Choose-Oracle Algorithm

Algorithm 3 is the full version of the Explore-then-Choose-Oracle algorithm.

---

**Algorithm 3** Explore-then-Choose-Oracle (Full version)

---

**Input:** $N$ workers, $K$ jobs, horizon $T$, exploration length $T_0 < T$, preference profile $P_a$ for all jobs $a \in \mathcal{K}$, approximation stable-matching oracle $\mathbb{O}$.

1: Initialize: $\hat{U}(i,j) = 0, T_{i,j} = 0, \forall i \in [N], j \in [K]$.
2: Initialize: $F_i \leftarrow$ False.            $\triangleright$ Whether the CIs of the first $(N+1)$-ranked jobs are disjoint.
3: Set $t = 1, T_0 \leftarrow K\lfloor T_0/K \rfloor, t_m = 0$            $\triangleright$ To have full rounds of round-robin.
4: **while** $t \leq T_0$ and $\exists\, i \in [N]$ s.t. $F_i ==$ False **do**            $\triangleright$ Phase 1, round-robin exploration.
5:     Match $\mu_t(i) \leftarrow a_{((t+i-1) \mod K)+1}, \forall\, i \in [N]$.
6:     Observe $X_{i,\mu_t(i)}(t)$ and update $\hat{U}(i, \mu_t(i)), T_{i,\mu_t(i)}$ as follows:

$$\hat{U}(i, \mu_t(i)) = \frac{\hat{U}(i, \mu_t(i)) \cdot T_{i,\mu_t(i)} + X_{i,\mu_t(i)}(t)}{T_{i,\mu_t(i)} + 1}, T_{i,\mu_t(i)} = T_{i,\mu_t(i)} + 1.$$

7:     $t \leftarrow t + 1$
8:     **if** $t \mod K == 0$ **then**            $\triangleright$ Completed a full round of round-robin
9:         $t_m \leftarrow t_m + 1$.
10:         Compute $UCB_{i,j}$ and $LCB_{i,j}$ for all $i \in [N], j \in [K]$.            $\triangleright$ See Equation (5)
11:         **for** $i = 1, 2, \cdots, N$ **do**
12:             $\hat{U}_{\text{sort}}(i, \cdot) \leftarrow \text{Sort}(\hat{U}(i, \cdot), \text{decreasing})$
13:             $\Delta_{i,\min} \leftarrow \min\left\{\hat{U}_{\text{sort}}(i,j) - \hat{U}_{\text{sort}}(i, j+1), j \in [N]\right\}$
14:             **if** $\Delta_{i,\min} > 2\sqrt{\frac{6 \ln T}{t_m}}$ **then**
15:                 $F_i \leftarrow$ True.
16:             **end if**
17:         **end for**
18:         **if** $F_i ==$ True, $\forall i \in [N]$, **then**            $\triangleright$ No ties – standard oracle
19:             Compute preference list $P_w$ for all $w \in \mathcal{W}$ according to $\hat{U}$
20:             $\hat{\mu}^* \leftarrow$ worker-optimal stable matching w.r.t $P_w$ and $P_a$ (using GS algorithm)
21:         **else if** $t = T_0$ **then**            $\triangleright$ Potential ties – approximation oracle
22:             $\hat{\mu}^* \leftarrow \mathbb{O}(\bar{U})$ for $\bar{U}$ s.t. $\bar{U}(i,j) = UCB_{i,j}$ for all $i \in [N], j \in [K]$
23:         **end if**
24:     **end if**
25: **end while**
26: **while** $t \leq T$ **do**            $\triangleright$ Phase 2, exploitation with the chosen oracle.
27:     Match $\mu_t(i) \leftarrow \hat{\mu}^*(i), \forall i$.
28:     $t \leftarrow t + 1$
29: **end while**

---

## F  Technical Lemmas

**Lemma 2** (Corollary 5.5 in Lattimore and Szepesvári [40])**.** *Assume that $X_1, X_2, \cdots, X_n$ are independent, $\sigma$-subgaussian random variables centered around $\mu$. Then for any $\varepsilon > 0$,*

$$\mathbb{P}\left(\frac{1}{n}\sum_{i=1}^n X_i \geq \mu + \varepsilon\right) \leq \exp\left(-\frac{n\varepsilon^2}{2\sigma^2}\right), \quad \mathbb{P}\left(\frac{1}{n}\sum_{i=1}^n X_i \leq \mu - \varepsilon\right) \leq \exp\left(-\frac{n\varepsilon^2}{2\sigma^2}\right).$$

**Lemma 3** (Divergence Decomposition, Lemma 15.1 in Lattimore and Szepesvári [40]). *For two bandit instances $\nu = \{\nu_{ij} : i \in [N], j \in [K]\}$, and $\nu' = \{\nu'_{ij} : i \in [N], j \in [K]\}$, fix some policy $\pi$ and let $\mathbb{P}_{\nu,\pi}$ and $\mathbb{P}_{\nu',\pi}$ be the probability measures induced by the $T$-round interconnection of $\pi$ and $\nu$ (respectively, $\pi$ and $\nu'$), the following divergence decomposition holds,*

$$D\left(\mathbb{P}_{\nu,\pi}, \mathbb{P}_{\nu',\pi}\right) = \sum_{i=1}^{N} \sum_{j=1}^{K} \mathbb{E}_{\nu,\pi} N_{ij}(T) \cdot D\left(\nu_{ij}, \nu'_{ij}\right). \tag{14}$$

**Lemma 4** (Data-processing Inequality, Lemma 1 in Garivier et al. [24]). *Consider a measurable space $(\Omega, \mathcal{F})$ equipped with two distributions $\mathbb{P}_1$ and $\mathbb{P}_2$, and any $\mathcal{F}$-measurable random variable $Z : \Omega \to [0,1]$. We denote respectively by $\mathbb{E}_1$ and $\mathbb{E}_2$ the expectations under $\mathbb{P}_1$ and $\mathbb{P}_2$. Then,*

$$KL\left(\mathbb{P}_1, \mathbb{P}_2\right) \geq kl(\mathbb{E}_1[Z], \mathbb{E}_2[Z]),$$

*where $kl$ denotes the KL divergence for Bernoulli distributions, i.e., $\forall p, q \in [0,1]^2, kl(p,q) = p \ln \frac{p}{q} + (1-p) \ln \frac{1-p}{1-q}$.*

## G   Proof of Therorem 7

For convenience, let $\hat{\boldsymbol{U}}^{(t)}(i,j)$, $T_{i,j}^{(t)}$, $UCB_{i,j}^{(t)}$, $LCB_{i,j}^{(t)}$ be the value of $\hat{\boldsymbol{U}}(i,j)$, $T_{i,j}$, $UCB_{i,j}$, $LCB_{i,j}$ at the end of round $t$. Define $\mathcal{F} = \left\{ \exists t \in [T], i \in [N], j \in [K] : |\hat{\boldsymbol{U}}^{(t)}(i,j) - \boldsymbol{U}(i,j)| > \sqrt{\frac{6 \ln T}{T_{i,j}^{(t)}}} \right\}$ as the bad event that some preference is not estimated well during the horizon.

**Lemma 5.**

$$\mathbb{P}(\mathcal{F}) \leq 2NK/T.$$

*Proof.*

$$\mathbb{P}(\mathcal{F}) = \mathbb{P}\left( \exists 1 \leq t \leq T, i \in [N], j \in [K] : |\hat{\boldsymbol{U}}^{(t)}(i,j) - \boldsymbol{U}(i,j)| > \sqrt{\frac{6 \ln T}{T_{i,j}^{(t)}}} \right)$$

$$\leq \sum_{t=1}^{T} \sum_{i \in [N]} \sum_{j \in [K]} \mathbb{P}\left( |\hat{\boldsymbol{U}}^{(t)}(i,j) - \boldsymbol{U}(i,j)| > \sqrt{\frac{6 \ln T}{T_{i,j}^{(t)}}} \right)$$

$$\leq \sum_{t=1}^{T} \sum_{i \in [N]} \sum_{j \in [K]} \sum_{s=1}^{t} \mathbb{P}\left( T_{i,j}^{(t)} = s, |\hat{\boldsymbol{U}}^{(t)}(i,j) - \boldsymbol{U}(i,j)| > \sqrt{\frac{6 \ln T}{s}} \right)$$

$$\leq \sum_{t=1}^{T} \sum_{i \in [N]} \sum_{j \in [K]} t \cdot 2 \exp(-3 \ln T)$$

$$\leq 2NK/T,$$

where the second last inequality results from Lemma 2. $\qquad \square$

**Lemma 6.** *Conditional on $\neg \mathcal{F}$, $UCB_{i,j}^{(t)} < LCB_{i,j'}^{(t)}$ implies $\boldsymbol{U}(i,j) < \boldsymbol{U}(i,j')$.*

*Proof.* According to the definition of LCB and UCB, we have that conditional on $\neg \mathcal{F}$,

$$LCB_{i,j}^{(t)} = \hat{\boldsymbol{U}}_{i,j}^{(t)} - \sqrt{\frac{6 \ln T}{T_{i,j}^{(t)}}} \leq \boldsymbol{U}(i,j) \leq \hat{\boldsymbol{U}}^{(t)}(i,j) + \sqrt{\frac{6 \ln T}{T_{i,j}^{(t)}}} = UCB_{i,j}^{(t)}.$$

Therefore, if $UCB_{i,j}^{(t)} < LCB_{i,j'}^{(t)}$, we have that

$$\boldsymbol{U}(i,j) \leq UCB_{i,j}^{(t)} \leq LCB_{i,j'}^{(t)} \leq \boldsymbol{U}(i,j').$$

$\qquad \square$

**Lemma 7.** *In round $t$, let $T_i^{(t)} = \min_{j \in [K]} T_{i,j}^{(t)}$. Conditional on $\neg \mathcal{F}$, if $T_i^{(t)} > 96 \ln T / \Delta_{\min}^2$, we have $LCB_{i,\rho_{i,k}}^{(t)} > UCB_{i,\rho_{i,k+1}}^{(t)}$ for any $k \in [N]$, and $LCB_{i,\rho_{i,N}}^{(t)} > UCB_{i,\rho i,k}^{(t)}$ for any $N+1 \le k \le K$.*

*Proof.* We prove it by contradiction, suppose that there exists $k \in [N]$ such that $LCB_{i,\rho_{i,k}}^{(t)} \le UCB_{i,\rho_{i,k+1}}^{(t)}$ or there exists $N+1 \le k \le K$ such that $LCB_{i,\rho_{i,N}}^{(t)} \le UCB_{i,\rho i,k}^{(t)}$. Without loss of generality, denote $j$ as the arm on the LHS and $j'$ as the arm on the RHS.

Conditional on $\neg \mathcal{F}$ and by the definition of $LCB$ and $UCB$, we have that

$$\boldsymbol{U}(i,j) - 2\sqrt{\frac{6 \ln T}{T_i^{(t)}}} \le LCB_{i,j}^{(t)} \le UCB_{i,j'}^{(t)} \le \boldsymbol{U}(i,j') + 2\sqrt{\frac{6 \ln T}{T_i^{(t)}}}.$$

Therefore, $\Delta_{i,j,j'} = \boldsymbol{U}(i,j) - \boldsymbol{U}_{i,j'} \le 4\sqrt{\frac{6 \ln T}{T_i^{(t)}}}$, which implies that $T_i^{(t)} \le \frac{96 \ln T}{\Delta_{i,j,j'}^2} \le \frac{96 \ln T}{\Delta_{\min}^2}$, which is a contradiction. $\square$

**Lemma 8.** *Conditional on $\neg \mathcal{F}$, if $\Delta_{\min} > \sqrt{\frac{96K \ln T}{T_0}}$, Algorithm 3 would enter the exploitation phase and choose the Gale-Shapley oracle at some $t \le T_0$.*

*Proof.* If $\Delta_{\min} > \sqrt{\frac{96K \ln T}{T_0}}$, we have $T_0 > \frac{96K \ln T}{\Delta_{\min}^2}$. Since for every worker, Algorithm 3 allocates jobs in a round-robin fashion, we have that $T_i^{(t)} > \frac{96 \ln T}{\Delta_{\min}^2}$.

By Lemma 7, we know that for any worker $w_i$, $LCB_{i,\rho_{i,k}}^{(t)} > UCB_{i,\rho_{i,k+1}}^{(t)}$ for any $k \in [N]$, and $LCB_{i,\rho_{i,N}}^{(t)} > UCB_{i,\rho i,k}^{(t)}$ for any $N+1 \le k \le K$, i.e., the preference utility for the first $N$-ranked jobs for every worker has been estimated well enough with the confidence intervals disjoint. The flag $F_i$ would be set as True as in Line 15 in Algorithm 3 and we would enter Phase 2 at some time $t \le T_0$. $\square$

**Lemma 9.** *Given a utility matrix $\boldsymbol{U}_{N \times K}$ without ties, the worker-optimal stable matching job of each worker must be its first $N$-ranked.*

*Proof.* We implement the Gale-Shapley algorithm with the workers as the proposing side. Once a job is proposed, it has a temporary worker. By contradiction, once $N$ jobs have been proposed, we have $N$ workers occupied. Therefore, each worker would be allocated with a job and the Gale-Shapley algorithm would stop. Since in the deferred-acceptance procedure, workers propose to jobs one by one according to their preference list, then the worker-optimal stable matching job of each worker must be its first $N$-ranked. $\square$

*Proof of Theorem 7.* We consider the two cases separately.

**Case 1.** $\Delta_{\min} > \sqrt{\frac{96K \ln T}{T_0}}$.

Let $\Delta_{i,\max} = \max_{j \in [K]} [\boldsymbol{U}^*(w_i) - \boldsymbol{U}(i,j)]$ be the maximum worker-optimal stable regret that may be suffered by $w_i$ in all rounds, we have $\Delta_{i,\max} \le 1$. The worker-optimal stable regret for each

worker $w_i$ by following Algorithm 3 satisfies

$$Reg_i(T) = \mathbb{E}\left[\sum_{t=1}^{T}\left(\boldsymbol{U}^*(w_i) - X_i(t)\right)\right]$$

$$\leq \mathbb{E}\left[\sum_{t=1}^{T} \mathbb{1}\{\mu_t(i) \neq \mu^*(i)\} \cdot \Delta_{i,\max}\right] \tag{15}$$

$$\leq \mathbb{E}\left[\sum_{t=1}^{T} \mathbb{1}\{\mu_t(i) \neq \mu^*(i)\} \mid \neg\mathcal{F}\right] \cdot \Delta_{i,\max} + \mathbb{P}(\mathcal{F}) \cdot T \cdot \Delta_{i,\max}$$

$$\leq \mathbb{E}\left[\sum_{t=1}^{T} \mathbb{1}\{\mu_t(i) \neq \mu^*(i)\} \mid \neg\mathcal{F}\right] \cdot \Delta_{i,max} + 2NK\Delta_{i,\max} \tag{16}$$

$$\leq \left\lceil \frac{96K\ln T}{\Delta_{\min}^2} \right\rceil \cdot \Delta_{i,max} + 2NK\Delta_{i,\max} \tag{17}$$

$$= O\left(\frac{K\ln T}{\Delta_{\min}^2}\right),$$

where Eq.(15) comes from the fact that in a matching market without ties, there is a unique worker-optimal stable matching and hence a unique optimal stable match $\mu^*(i)$ for worker $i$, Eq.(16) holds based on Lemma 5, Eq.(17) holds according to Lemma 8 and 9 and the fact that Gale-Shapley algorithm could always output the worker-optimal stable matching with respect to the given utility matrix by treating worker as the proposing side.

**Case 2.** $\Delta_{\min} \leq \sqrt{\frac{96K\ln T}{T_0}}$.

The objective function is the approximation regret $Reg_i^\alpha(T)$. Denote $\mathcal{F}_d^{(t)}$ as the event that $LCB_{i,\rho_{i,k}}^{(t)} > UCB_{i,\rho_{i,k+1}}^{(t)}$ for all $k \in [N]$, and $LCB_{i,\rho_{i,N}}^{(t)} > UCB_{i,\rho i,k}^{(t)}$ for all $N+1 \leq k \leq K$. We have

$$Reg_i^\alpha(T) = \mathbb{E}\left[\alpha T \cdot \boldsymbol{U}^*(w_i) - \sum_{t=1}^{T} X_i(t) \mid \mathcal{F}\right] \cdot \mathbb{P}(\mathcal{F})$$

$$+ \mathbb{E}\left[\alpha T \cdot \boldsymbol{U}^*(w_i) - \sum_{t=1}^{T} X_i(t) \mid \neg\mathcal{F}\right] \cdot \mathbb{P}(\neg\mathcal{F})$$

$$\leq \alpha T \cdot \mathbb{P}(\mathcal{F}) + \mathbb{E}\left[\alpha T \cdot \boldsymbol{U}^*(w_i) - \sum_{t=1}^{T} X_i(t) \mid \neg\mathcal{F}\right] \tag{18}$$

$$\leq 2\alpha NK + \mathbb{E}\left[\alpha T \cdot \boldsymbol{U}^*(w_i) - \sum_{t=1}^{T} X_i(t) \mid \neg\mathcal{F}\right] \tag{19}$$

$$\leq 2\alpha NK + \mathbb{E}\left[\left(\alpha T \cdot \boldsymbol{U}^*(w_i) - \sum_{t=1}^{T} X_i(t)\right) \mathbb{1}\left\{\mathcal{F}_d^{(T_0)}\right\} \mid \neg\mathcal{F}\right]$$

$$+ \mathbb{E}\left[\left(\alpha T \cdot \boldsymbol{U}^*(w_i) - \sum_{t=1}^{T} X_i(t)\right) \mathbb{1}\left\{\neg\mathcal{F}_d^{(T_0)}\right\} \mid \neg\mathcal{F}\right]$$

$$\leq 2\alpha NK + \alpha T_0 + \mathbb{E}\left[\left(\alpha T \cdot \boldsymbol{U}^*(w_i) - \sum_{t=1}^{T} X_i(t)\right) \mathbb{1}\left\{\neg\mathcal{F}_d^{(T_0)}\right\} \mid \neg\mathcal{F}\right], \tag{20}$$

where Eq.(18) comes from the fact that $\boldsymbol{U}^*(w_i) \leq 1$ and $X_i(t) \geq 0$. Eq.(19) holds according to Lemma 5. Eq.(20) comes from the fact that when the good event $\neg\mathcal{F}$ that all utilities are well estimated and the top $(N+1)$-ranked CIs are disjoint before $T_0$, the Gale-Shapley algorithm would give us the OSS in the exploitation phase, and since $\alpha \in (0,1]$, the approximation regret would be no larger than $\alpha T_0 + (\alpha - 1) \cdot (T - T_0) \leq \alpha T_0$.

Moreover, conditional on $\neg\mathcal{F}$, we have the ground-truth utility matrix $\boldsymbol{U}$ lies in the uncertainty set constructed by the empirical mean utility matrix $\hat{\boldsymbol{U}}^{(T_0)}$ and the $UCB^{(T_0)}$ and $LCB^{(T_0)}$, i.e., for any $(i,j) \in [N] \times [K]$, $|\hat{\boldsymbol{U}}^{(T_0)}(i,j) - \boldsymbol{U}(i,j)| \leq \sqrt{\frac{6K\ln T}{T_0}}$. If we implement an $(\boldsymbol{\alpha}, \epsilon)$-oracle, with $\epsilon$ being $2\sqrt{\frac{6K\ln T}{T_0}}$, follow a similar proof as that for Theorem 6, in each round $t$ in the exploitation phase, we have that

$$\alpha \boldsymbol{U}^*(w_i) - \mathbb{E}X_i(t) \leq 2\sqrt{\frac{6K\ln T}{T_0}}. \tag{21}$$

Since there are in total $T - T_0$ rounds of exploitation, we have that

$$\mathbb{E}\left[\left(\alpha T \cdot \boldsymbol{U}^*(w_i) - \sum_{t=1}^{T} X_i(t)\right) \mathbb{1}\left\{\neg\mathcal{F}_d^{(T_0)}\right\} \mid \neg\mathcal{F}\right] \leq \alpha T_0 + 2\sqrt{\frac{6K\ln T}{T_0}}(T - T_0). \tag{22}$$

Therefore, combining Eq.(20) and (22), we have

$$Reg_i^\alpha(T) \leq 2\alpha NK + 2\alpha T_0 + 2\sqrt{\frac{6K\ln T}{T_0}}(T - T_0).$$

$\square$

## H    Proof of Theorem 8

*Proof.* Let $\mathcal{W} = \{w_1, w_2, w_3, w_4\}$ and $\mathcal{A} = \{a_1, a_2, a_3, a_4\}$ and $w_1 \succ w_2 \succ w_3 \succ w_4$ for all the jobs. Throughout the proof, we assume that all observations are Gaussian of unit variance, that is, when matching $w_i$ to $a_j$ at round $t$, we observe $X_i(t) \sim \mathcal{N}(\boldsymbol{U}(i,j), 1)$. Consider two instances $\boldsymbol{\nu}$ and $\boldsymbol{\nu}'$ with the following mean utility matrices $\boldsymbol{U}$ and $\boldsymbol{U}'$, respectively.

$$\boldsymbol{U} = \begin{bmatrix} \frac{1}{2} & \frac{1}{2} & 0 & 0 \\ \frac{1}{2} & 0 & \frac{1}{2} & 0 \\ \frac{1}{2} & 0 & 0 & \frac{1}{4} \\ 0 & 0 & \frac{1}{2} & 0 \end{bmatrix}, \quad \boldsymbol{U}' = \begin{bmatrix} \frac{1}{2}+\gamma & \frac{1}{2} & 0 & 0 \\ \frac{1}{2} & 0 & \frac{1}{2} & 0 \\ \frac{1}{2} & 0 & 0 & \frac{1}{4} \\ 0 & 0 & \frac{1}{2} & 0 \end{bmatrix},$$

where $\gamma < \frac{1}{4}$.

**Lemma 10** (Properties of Instances $\boldsymbol{\nu}$ and $\boldsymbol{\nu}'$). *Based on the utility matrices $\boldsymbol{U}$ and $\boldsymbol{U}'$, we have the following properties of $\boldsymbol{\nu}$ and $\boldsymbol{\nu}'$:*

1. *Under $\boldsymbol{\nu}$, the optimal stable shares are $\boldsymbol{U}^*(w) = \frac{1}{2}, \forall w \in \mathcal{W}$; Under $\boldsymbol{\nu}'$, the optimal stable shares are $(\boldsymbol{U}')^*(w_1) = \frac{1}{2} + \gamma$, $(\boldsymbol{U}')^*(w_2) = \frac{1}{2}$, $(\boldsymbol{U}')^*(w_3) = \frac{1}{4}$, and $(\boldsymbol{U}')^*(w_4) = 0$.*

2. *The relevant utility gaps for the two instances are $\Delta_{rel}^{\boldsymbol{\nu}} = 0$, and $\Delta_{rel}^{\boldsymbol{\nu}'} = \gamma$.*

3. *Given an offline oracle that could compute the best approximation ratio, the benchmark utilities for the four workers (after multiplying the approximation ratio) are $(\frac{1}{2}, \frac{3}{8}, \frac{3}{8}, \frac{3}{8})$ under $\boldsymbol{\nu}$ and $(\frac{1}{2} + \gamma, \frac{1}{2}, \frac{1}{4}, 0)$ under $\boldsymbol{\nu}'$.*

We provide the proof of Lemma 10 in Appendix I.

For a worker $w_i, i \in [N]$, job $a_j, j \in [K]$ and time slot $t \in [T]$, denote $N_{ij}(t) \in \mathbb{N} \cup \{0\}$ as the number of times worker $w_i$ is matched to job $a_j$, up to and including time $t$, and denote the past information as $I_t := (\mu_1, \boldsymbol{X}(1), \mu_2, \boldsymbol{X}(2), \cdots, \mu_{t-1}, \boldsymbol{X}(t-1))$, where $\boldsymbol{X}(t) = (X_1(t), X_2(t), \cdots, X_N(t))$ is the realized reward vector for all $N$ workers in round $t$. Finally, let $\mathbb{P}_{\boldsymbol{\nu},\pi}$ be the joint probability measure over the history and $\mathbb{E}_{\boldsymbol{\nu},\pi}$ be the expectation induced by instance $\boldsymbol{\nu}$ and policy $\pi$, and $\mathbb{P}_{\boldsymbol{\nu}',\pi}$, $\mathbb{E}_{\boldsymbol{\nu}',\pi}$ be defined similarly. By divergence decomposition theorem [40, restated in Lemma 3], we have that

$$D_{\mathrm{KL}}\left(\mathbb{P}_{\boldsymbol{\nu},\pi}, \mathbb{P}_{\boldsymbol{\nu}',\pi}\right) = \sum_{i=1}^{N}\sum_{j=1}^{K} \mathbb{E}_{\boldsymbol{\nu},\pi} N_{ij}(T) \cdot D_{\mathrm{KL}}\left(\boldsymbol{\nu}_{ij}, \boldsymbol{\nu}'_{ij}\right),$$

where $\boldsymbol{\nu}_{ij}$ is the distribution of utilities obtained when worker $w_i$ is matched to job $a_j$ in the environment $\boldsymbol{\nu}$.

Since the only change in utility distribution happens in $(w_1, a_1)$ pair, we have that

$$D_{\mathrm{KL}}\left(\mathbb{P}_{\boldsymbol{\nu},\pi}, \mathbb{P}_{\boldsymbol{\nu}',\pi}\right) = D_{\mathrm{KL}}(\boldsymbol{\nu}_{11}, \boldsymbol{\nu}'_{11})\mathbb{E}_{\boldsymbol{\nu},\pi}\left[N_{11}(T)\right] = \mathbb{E}_{\boldsymbol{\nu},\pi}\left[N_{11}(T)\right] \cdot \frac{\gamma^2}{2}, \tag{23}$$

where the second equality comes from the fact that for two Gaussian distributions with means $\frac{1}{2}$ and $\frac{1}{2} + \gamma$ and variance 1, the KL divergence is $\frac{\gamma^2}{2}$.

By data-processing inequality [40, restated in Lemma 4], we know that for all $\sigma(I_T)$-measurable random variable $Z \in [0, 1]$, we have that

$$D_{\mathrm{KL}}\left(\mathbb{P}_{\boldsymbol{\nu},\pi}, \mathbb{P}_{\boldsymbol{\nu}',\pi}\right) \geq \mathrm{kl}\left(\mathbb{E}_{\boldsymbol{\nu},\pi}(Z), \mathbb{E}_{\boldsymbol{\nu}',\pi}(Z)\right). \tag{24}$$

where $\mathrm{kl}$ denotes the KL divergence between two Bernoulli distributions, i.e., $\forall p, q \in [0,1]^2, \mathrm{kl}(p,q) = p\ln\frac{p}{q} + (1-p)\ln\frac{1-p}{1-q}$.

Let $Z = \frac{N_{21}(T) + N_{23}(T)}{T}$, then $Z \in [0, 1]$, by Pinsker's inequality, we have that

$$\mathrm{kl}\left(\mathbb{E}_{\boldsymbol{\nu},\pi}(Z), \mathbb{E}_{\boldsymbol{\nu}',\pi}(Z)\right) \geq 2 \cdot \left[\mathbb{E}_{\boldsymbol{\nu},\pi}(Z) - \mathbb{E}_{\boldsymbol{\nu}',\pi}(Z)\right]^2. \tag{25}$$

Combining Eq.(23), (24), (25), we have that

$$\mathbb{E}_{\boldsymbol{\nu},\pi}\left[N_{11}(T)\right] \cdot \frac{\gamma^2}{2} \geq 2 \cdot \left[\mathbb{E}_{\boldsymbol{\nu},\pi}(Z) - \mathbb{E}_{\boldsymbol{\nu}',\pi}(Z)\right]^2. \tag{26}$$

We now divide into two cases, depending on the asymptotic number of matches $\mathbb{E}_{\boldsymbol{\nu},\pi}[N_{11}(T)]$.

**Case I:** $\liminf_{T\to\infty}\frac{\mathbb{E}_{\boldsymbol{\nu},\pi}[N_{11}(T)]}{T^{1-2\delta}} = 0$. We assume that both $Reg_i(T; \boldsymbol{\nu}')$ and $Reg_i^{\boldsymbol{\alpha}^*(w_i)}(T; \boldsymbol{\nu})$ are sublinear for all workers and show that we have a contradiction.

Since $\gamma = cT^{-\frac{1}{2}+\delta}$, by Eq. (26), we have

$$\liminf_{T\to\infty}\frac{|\mathbb{E}_{\boldsymbol{\nu},\pi}[N_{21}(T) + N_{23}(T)] - \mathbb{E}_{\boldsymbol{\nu}',\pi}[N_{21}(T) + N_{23}(T)]|}{T} = 0. \tag{27}$$

In $\boldsymbol{\nu}'$, if the ground-truth utility matrix $\boldsymbol{U}'$ is known and we would like to achieve the benchmark utility for $w_2$, we need $N_{21}(T) + N_{23}(T) = T$, since worker $w_2$ could only get positive utilities from jobs $a_1$ and $a_3$ and her benchmark utility is $\frac{1}{2}$. In particular, the regret for this worker is

$$Reg_i(T; \boldsymbol{\nu}') = \frac{1}{2}(T - \mathbb{E}_{\boldsymbol{\nu}',\pi}[N_{21}(T) + N_{23}(T)]),$$

and to guarantee sublinear regret for $w_2$ for any large enough $T$, we must have

$$\liminf_{T\to\infty}\frac{\mathbb{E}_{\boldsymbol{\nu}',\pi}\left[N_{21}(T) + N_{23}(T)\right]}{T} = 1.$$

Therefore, to satisfy Eq.(27), we must also have

$$\liminf_{T\to\infty}\frac{\mathbb{E}_{\boldsymbol{\nu},\pi}\left[N_{21}(T) + N_{23}(T)\right]}{T} = 1. \tag{28}$$

On the other hand, since $w_4$ only gets positive utilities in $\boldsymbol{U}(4, 3)$, to achieve the benchmark utility in $\boldsymbol{\nu}$, we need $N_{43}(T) = \frac{3}{4}T$. Therefore, to guarantee sublinear approximation regret for $w_4$, we must have

$$\liminf_{T\to\infty}\frac{\mathbb{E}_{\boldsymbol{\nu},\pi}\left[N_{43}(T)\right]}{T} \geq \frac{3}{4},$$

which implies $\limsup_{T\to\infty}\frac{\mathbb{E}_{\boldsymbol{\nu},\pi}[N_{23}(T)]}{T} \leq \frac{1}{4}$, since the total number of times that jobs $a_3$ being allocated cannot be more than the horizon $T$. Therefore, $\liminf_{T\to\infty}\frac{\mathbb{E}_{\boldsymbol{\nu},\pi}[N_{21}(T)]}{T} \geq \frac{3}{4}$ according to Eq.(28). Using again the fact that a job can be allocated no more than $T$ times, we get

$$\limsup_{T\to\infty}\frac{\mathbb{E}_{\boldsymbol{\nu},\pi}\left[N_{31}(T)\right]}{T} \leq \frac{1}{4}. \tag{29}$$

Finally, we write the regret of $w_3$ as

$$
\begin{aligned}
Reg_3^{\alpha}(T) &= \frac{3T}{8} - \frac{1}{2}\mathbb{E}_{\boldsymbol{\nu},\pi}[N_{31}(T)] - \frac{1}{4}\mathbb{E}_{\boldsymbol{\nu},\pi}[N_{34}(T)] \\
&= \frac{3T}{8} - \frac{1}{4}\mathbb{E}_{\boldsymbol{\nu},\pi}[N_{31}(T)] - \frac{1}{4}\left(\mathbb{E}_{\boldsymbol{\nu},\pi}[N_{34}(T)] + \mathbb{E}_{\boldsymbol{\nu},\pi}[N_{31}(T)]\right) \\
&\geq \frac{T}{8} - \frac{1}{4}\mathbb{E}_{\boldsymbol{\nu},\pi}[N_{31}(T)],
\end{aligned}
$$

where the inequality is since $w_3$ is matched at most $T$ times, namely, $\mathbb{E}_{\boldsymbol{\nu},\pi}[N_{34}(T)] + \mathbb{E}_{\boldsymbol{\nu},\pi}[N_{31}(T)] \leq T$. Combining with Eq. (29), we have $\liminf_{T\to\infty} \frac{Reg_3^{\alpha}(T;\boldsymbol{\nu},\pi)}{T} \geq \frac{1}{8} - \frac{1}{4} \cdot \frac{1}{4} = \frac{1}{16}$, which implies worker $w_3$ suffers linear approximation regret in $\boldsymbol{\nu}$.

Thus, to summarize, assuming that all workers in both problem exhibit sublinear regret for this case leads to a contradiction.

**Case II:** $\liminf_{T\to\infty} \frac{\mathbb{E}_{\boldsymbol{\nu},\pi}[N_{11}(T)]}{T^{1-2\delta}} > 0$. Our goal is to prove a lower bound on the regret in $\boldsymbol{\nu}$ for some worker.

For a fixed $T$, denote for brevity $N = \mathbb{E}_{\boldsymbol{\nu},\pi}[N_{11}(T)]$, and assume with contradiction that all workers $w_2, w_3, w_4$ suffer a regret smaller than $N/32$. Denote the cumulative allocation given by the algorithm by $D \in [0,T]^{4\times 4}$, namely $D(i,j) = \sum_{t=1}^{T} \mathbb{1}\{(w_i, a_j) \in \mu_t\}$. In particular, we know that $D(1,1) = N$ and that for all $i \in \{2,3,4\}$, it holds that

$$
\forall i \in \{2,3,4\}, \quad \sum_{j=1}^{4} \boldsymbol{U}(i,j)D(i,j) \geq \frac{3T}{8} - \frac{N}{32} \tag{30}
$$

We now state a set of assumptions on the matching that the policy outputs at each round. Each of these assumptions never decreases the worker utility. Thus, and since we want to prove a contradiction in the utility lower bound of Eq. (30), they could be assumed without loss of generality.

1. When $w_1$ is not matched to $a_1$, it is always matched to $a_2$ ($D(1,1) + D(1,2) = T$) – so that it suffers zero regret.

2. $a_3$ is always matched to either $w_2$ or $w_4$ ($D(2,3) + D(4,3) = T$).

3. $a_1$ is always assigned to one of the first three workers ($D(1,1) + D(2,1) + D(3,1) = T$).

4. If $a_1$ is not assigned to $w_3$, then $a_4$ is assigned to $w_3$ ($D(3,1) + D(3,4) = T$).

Notice that changing each individual allocation to follow this condition can only require unmatching a worker from a job that yields her no utility and matching all conditions is feasible (e.g., by $\mu = \{(w_1, a_1), (w_3, a_4), (w_4, a_3)\}$).

We now modify the matching allocation $D$ to allocation $\bar{D}$ while maintaining the above properties as follows:

- We initialize $\bar{D} = D$.

- If $D(4,3) \leq \frac{3T}{4} + \frac{N}{16}$, we set $\bar{D}(4,3) = \frac{3T}{4} + \frac{N}{16}$ and $\bar{D}(2,3) = \frac{T}{4} - \frac{N}{16}$; otherwise, we leave $\bar{D}(4,3) = D(4,3)$. By Eq. (30), we know that $D(4,3) \geq \frac{3T}{4} - \frac{N}{16}$, and combined with Assumption 2, this change can only decrease the allocation to worker $w_2$ by

$$
D(2,3) - \bar{D}(2,3) = \bar{D}(4,3) - D(4,3) \leq \frac{N}{8}
$$

This decreases the utility of worker $w_2$ by $\frac{1}{2}\left(D(2,3) - \bar{D}(2,3)\right) \leq \frac{N}{16}$, and after this modification, the cumulative utility of worker $w_4$ is $\frac{3T}{8} + \frac{N}{32}$.

- By Assumption 1, we know that $D(1,1) = N$ and $D(1,2) = T - N$. We also know by assumption 4 that in all rounds where $a_1$ was assigned to $w_1$, $a_4$ was assigned to $w_3$, and therefore, $D(3,4) \geq N$. In $\bar{D}$, we move all the $N$ assignments of $(w_1, a_1)$ to $(w_1, a_2)$, so that $\bar{D}(1,1) = 0$ and $\bar{D}(1,2) = T$; in particular, $w_1$ still gets its OSS. We split the allocation of $a_1$ evenly between $w_2$ and $w_3$ by letting:

1. $\bar{D}(2,1) = \min\{T - \bar{D}(2,3), D(2,1) + N/2\}$, thus making sure that the utility of $w_2$ is at least either $T/2 \geq 3T/8 + N/16$ or

$$\frac{1}{2}\left(\bar{D}(2,1) + \bar{D}(2,3)\right) \geq \frac{1}{2}\left(D(2,1) + \frac{N}{2} + D(2,3) - \frac{N}{8}\right) \geq \frac{3T}{8} - \frac{N}{32} + \frac{3N}{16} = \frac{3T}{8} + \frac{5N}{32},$$

where in the last inequality we again used the assumption on the regret of $w_2$ in Eq.(30).

2. We move the matches from $(w_1, a_1)$ that were not allocated to $D(2,1)$ as follows:

$$\bar{D}(3,1) = D(3,1) + N - \left(\bar{D}(2,1) - D(2,1)\right) \geq D(3,1) + N/2, \quad \text{and,}$$
$$\bar{D}(3,4) = D(3,4) - \left(N - \left(\bar{D}(2,1) - D(2,1)\right)\right) \geq D(3,4) - N/2.$$

Both allocations are valid since $D(3,4) \geq N$ due to Assumption 4. In particular, this shift from $a_4$ to $a_1$ increases the utility of $w_3$ by at least $\left(\frac{1}{2} - \frac{1}{4}\right)\frac{N}{2} = \frac{N}{8}$, ensuring it a total utility of at least $3T/8 + 3N/32$.

Notice that all changes either kept $\bar{D}$ a doubly-stochastic matrix or decreased the sum of a row - we can w.l.o.g increase another element of $\bar{D}$ or use partial matchings.

Importantly, at the end of this process, the utility of $w_1$ under the matching distribution $\bar{D}$ remained $T/2$, while the utility of all other workers increased by at least $N/16$ - contradicting the fact that no matching distribution can collect more than $3T/8$ to all workers $w_2, w_3, w_4$. Thus, Eq. (30) cannot hold and at least one worker must suffer a regret of at least $N/32$. Finally, since $\liminf_{T\to\infty} \frac{\mathbb{E}_{\nu,\pi}[N_{11}(T)]}{T^{1-2\delta}} > 0$, we get the same for the regret of one of the workers, concluding the proof. $\qquad\square$

# I  Proof of Lemma 10

*Proof.* We proof the three properties as follows.

**1. Optimal Stable Share**

Under $\nu$, consider the following stable matchings: $\mu_1 = \{(w_1, a_2), (w_2, a_1), (w_3, a_4), (w_4, a_3)\}$ and $\mu_2 = \{(w_1, a_2), (w_2, a_3), (w_3, a_1), (w_4, a_4)\}$. The optimal stable share is $U^*(w) = \frac{1}{2}, \forall w \in \mathcal{W}$ with $w_1$ and $w_2$ receives it in both matchings, $w_3$ receives it in matching $\mu_2$ and $w_4$ receives it in matching $\mu_1$. Under $\nu'$, the optimal stable shares are $(u')^*(w_1) = \frac{1}{2} + \gamma$, $(u')^*(w_2) = \frac{1}{2}$, $(u')^*(w_3) = \frac{1}{4}$, and $(u')^*(w_4) = 0$, achieved through the stable matching $\mu' = \{(w_1, a_1), (w_2, a_3), (w_3, a_4), (w_4, a_2)\}$.

**2. Relevant Utility Gap**

Besides, the relevant utility gap for $\nu$ is $\Delta_{\text{rel}}^{\nu} = 0$ since both $\mu_1$ and $\mu_2$ belongs to the Pareto-optimal stable matching set $\mathcal{S}_{opt}^{U}$. On the other hand, since the jobs have global preference rankings over the workers, i.e., serial dictatorship, $\mu'$ is the unique stable matching with respect to $\nu'$, i.e., $\mathcal{S}_{opt}^{U'} = \{\mu'\}$. A perturbation of $-\gamma$ in $U(1,1)$ or $\gamma$ in $U(1,2)$ brings ties for worker $w_1$, and hence would change $\mathcal{S}_{opt}^{U'}$ tp $\mathcal{S}_{opt}^{U}$. On the other hand, a perturbation of $\frac{1}{4}$ in $U(3,2)$ or $-\frac{1}{4}$ in $U(3,4)$ would make $\mathcal{S}_{opt}^{U'}$ include both $\mu'$ and $\mu_2' = \{(w_1, a_1), (w_2, a_3), (w_3, a_2), (w_4, a_4)\}$. All other entries need a perturbation of scale larger than $\frac{1}{4}$ to change the Pareto-optimal stable matching set. Since $\gamma < \frac{1}{4}$, we have that $\Delta_{\text{rel}}^{\nu'} = \gamma$.

**3. Benchmark Utility**

Let $\gamma = cT^{-1/2+\delta}$ for some $\delta \in (0, \frac{1}{2})$. Then, under $\nu$, we aim to minimize the approximation regret $Reg_i^\alpha(T)$ for every worker $w_i$, while under $\nu'$, the objective is to minimize regret $Reg_i(T)$ for each worker $w_i$. Given an offline oracle that could compute the best possible approximation ratio, the benchmark utilities for the four workers (after multiplying the approximation ratio) are $\left(\frac{1}{2}, \frac{3}{8}, \frac{3}{8}, \frac{3}{8}\right)$ under $\nu$ and $\left(\frac{1}{2} + \gamma, \frac{1}{2}, \frac{1}{4}, 0\right)$ under $\nu'$. For $\nu'$, by serial dictatorship, the allocation scheme is equivalent to letting the workers choose their favorite jobs one by one. Doing so leads to the matching $\mu = \{(w_1, a_2), (w_2, a_3), (w_3, a_1), (w_4, a_4)\}$, which is unique since for all workers, when they choose their jobs, only a single unallocated job maximizes their utility. Hence, the benchmark utilities are immediately determined by the utility under this matching.

For $\boldsymbol{\nu}$, since all workers but $w_1$ have no utility from job $a_2$, and $\boldsymbol{U}^*(w_1) = \boldsymbol{U}(1, 2)$, it is always optimal to assign $a_2$ to $w_1$ deterministically and the benchmark utility for $w_1$ is $1/2$. For the other players, if we match $w_2$ to $a_1$, then for $w_3$ and $w_4$, there are two possible matchings, i.e., $\{(w_3, a_4), (w_4, a_3)\}$ and $\{(w_3, a_3), (w_4, a_4)\}$, but the first one is always better since it gives both players higher utilities. Similarly, if we match $w_2$ to $a_3$, it is always better to select the matching $\{(w_3, a_1), (w_4, a_4)\}$rather than $\{(w_3, a_4), (w_4, a_3)\}$, and if $w_2$ is matched to $a_4$, then we choosing $\{(w_3, a_1), (w_4, a_3)\}$ yields higher utilities than choosing $\{(w_3, a_3), (w_4, a_1)\}$. Since all three three players have an OSS of $1/2$, to maximize the OSS ratio, we need to compute a distribution $D$ over the three matchings $\mu_a = \{(w_2, a_1), (w_3, a_4), (w_4, a_3)\}$, $\mu_b = \{(w_2, a_3), (w_3, a_1), (w_4, a_4)\}$ and $\mu_c = \{(w_2, a_4), (w_3, a_1), (w_4, a_3)\}$, such that $\min\{u_D(w_2), u_D(w_3), u_D(w_4)\}$ is maximized. Noticing that each of the three matchings yields the OSS to two players and a lower utility for the third one, we can conclude that the optimal balance would be $u_D(w_2) = u_D(w_3) = u_D(w_4)$ – otherwise, we could increase the OSS-ratio by moving utility from the highest rewarded player to the lowest rewarded one. This condition is satisfied iff

$$\mathbb{P}(\mu_a) = \frac{1}{2}, \quad \mathbb{P}(\mu_b) = \frac{1}{4}, \quad \mathbb{P}(\mu_c) = \frac{1}{4},$$

and the benchmark utility for any of these three players is $\frac{3}{8}$. $\qquad\square$

## J  Discussion Regarding Stable Matching with One-sided Ties and Job Utilities

In this paper, we consider a matching market where one side has possibly tied cardinal preferences and the other side has strict ordinal preferences. It directly generalizes to the setting that both sides have cardinal preferences but only one side admits ties, by recovering an ordinal preference list from the utility, and we can define the OSS-ratio for the jobs in a similar fashion, denoted as $R_{\mathcal{M}}^a$; in the following, we rename the OSS-ratio for the workers as $R_{\mathcal{M}}^w$ for distinguishment. We claim that the setting with one-sided ties is not only practical in reality, but also important for our theoretical results, if we want to consider the OSS-ratio for both sides of the market.

**Stable Matching Without Ties**  The distributive lattice structure is a striking feature for a matching market without ties [36], which reveals that all workers could be optimally matched simultaneously, by simply running the deferred-acceptance algorithm with worker proposing, denoted as $\mu_w$. Conversely, job-proposing deferred-acceptance algorithm, denoted as $\mu_a$, gives every job its corresponding optimal stable match. Therefore, construct the distribution $D$ as follows:

$$\mathbb{P}(D = \mu_w) = \frac{1}{2}, \quad \mathbb{P}(D = \mu_a) = \frac{1}{2}.$$

Since $\mu_w$ and $\mu_a$ are both stable matchings, with distribution $D$, we know that $R_{\mathcal{M}}^w \leq R_{\mathcal{S}}^w \leq \frac{1}{2}$, and the same result holds for $R_{\mathcal{M}}^a$. This implies that both $R_{\mathcal{M}}^w$ and $R_{\mathcal{M}}^a$ are $\Theta(1)$, and $\Omega(1)$ is a trivial lower bound for the two ratios.

**Stable Matching With One-sided Ties**  The lattice structure is absent when ties exist in the preference profiles [45]. When only one side of the market admits ties, we have proved that $R_{\mathcal{M}}^w = \Theta(\log N)$. On the other hand, for $R_{\mathcal{M}}^a$, we have the following result.

**Theorem 9.** *For a matching market where the workers have ties while the jobs have strict preferences, there exists an instance such that $R_{\mathcal{M}}^a = \Omega(N)$.*

*Proof.* Consider the following utility matrix for a market with $N$ workers and $N$ jobs. The matrix encodes the preferences of the workers and the jobs simultaneously.

$$\boldsymbol{U} = \begin{bmatrix} 1 & 1 & \cdots & 1 \\ \varepsilon_1 & \varepsilon_1 & \cdots & \varepsilon_1 \\ \varepsilon_2 & \varepsilon_2 & \cdots & \varepsilon_2 \\ \vdots & \vdots & \ddots & \vdots \\ 0 & 0 & \cdots & 0 \end{bmatrix},$$

where $\varepsilon_1 > \varepsilon_2 > \cdots > 0$. $u_{i,j} \geq u_{i,j'}$ implies $a_j \succsim_{w_i} a_{j'}$ and $u_{i,j} \geq u_{i',j}$ implies $w_i \succsim_{a_j} w_{i'}$. In this example, every worker is indifferent among all the jobs, while every job has the preference profile that $w_1 \succ w_2 \succ \cdots \succ w_N$. The optimal stable share for each job is 1, and it is achieved by an appropriate tie-breaking of $w_1$. However, in any matching, exactly one job would receive a utility of 1. When $\varepsilon_1, \varepsilon_2, \cdots$ approach 0, we have that $\max_a \frac{\boldsymbol{U}^*(a)}{\boldsymbol{U}_D(a)} \geq N$, with the equality achieved when we consider the distribution $D$ that assigns probability $1/N$ on matching $\mu_j$, where $\mu_j$ refers to a matching in which job $a_j$ gets a utility of 1. Therefore, $\lim_{\varepsilon \to 0} R_{\mathcal{M}}^a = \Omega(N)$. $\qquad\square$

**Stable Matching with Two-sided Ties**     When both sides of the market have ties, by symmetry, the OSS-ratio for the worker side and the job side would be of the same order.

**Theorem 10.** *For a matching market where both sides admit ties, there exists an instance, such that $R_{\mathcal{M}}^w = R_{\mathcal{M}}^a = \Omega(N)$.*

*Proof.* Consider the following $N \times N$ utility matrix which simultaneously encodes the preferences of the workers and the jobs.

$$\boldsymbol{U} = \begin{bmatrix} 1 & 1 & \cdots & 1 \\ 1 & 0 & \cdots & 0 \\ \vdots & \vdots & \ddots & \vdots \\ 1 & 0 & \cdots & 0 \end{bmatrix},$$

where the entries of this utility matrix share the same correspondence to the preference profile as those in the example in Theorem 9. In this example, worker $w_1$ is indifferent among all the jobs, while all the other workers share the same preference over the jobs, that is, $a_1 \succ a_2 \sim a_3 \sim \cdots \sim a_N$. The preference of jobs over workers is symmetrically derived. Every matching that involves all the workers and jobs is a stable matching, which gives a utility of 1 to worker $w_1$ and job $a_1$, while for the remaining workers and jobs, at most one worker and one job would receive a utility of 1, and all the other workers get 0. For every worker and every job, there exists a tie-breaking mechanism such that it gets a utility of 1. Therefore, the optimal stable share is $\boldsymbol{U}^*(w) = \boldsymbol{U}^*(a) = 1$ for any $w$ and $a$. And any distribution $D$ over matchings gives $\frac{\boldsymbol{U}^*(w_1)}{\boldsymbol{U}_D(w_1)} = \frac{\boldsymbol{U}^*(a_1)}{\boldsymbol{U}_D(a_1)} = 1$, $\max_{w \in \{w_2, w_3, \cdots, w_N\}} \frac{\boldsymbol{U}^*(w)}{\boldsymbol{U}_D(w)} \geq N - 1$, and $\max_{a \in \{a_2, a_3, \cdots, a_N\}} \frac{\boldsymbol{U}^*(a)}{\boldsymbol{U}_D(a)} \geq N - 1$, with the equality achieved when we adopt the random allocation that assigns probability $1/(N-1)$ on matching $\mu_i$, in which workers $w_1$ and $w_i$, jobs $a_1$ and $a_i$ receive a utility of 1, while all the other workers and jobs get 0. $\qquad\square$

# K    Discusssion Regarding Two-sided Bandit Learning in Matching Markets

In this paper, we consider bandit learning for one side of the market, where the preferences of jobs over workers are assumed to be known. A possible future direction would be to consider two-sided bandit learning for matching markets.

For a matching market without ties, a fundamental result indicates that the worker-optimal stable matching is necessarily job-pessimal, and no stable matching simultaneously maximizes utility for both sides. Therefore, a reasonable benchmark would be to consider the optimal stable matching for workers and the pessimal stable matching for jobs, leading to the following regret definition:

$$Reg_i(T) = T \cdot \bar{\boldsymbol{U}}(w_i) - \mathbb{E}\left[\sum_{t=1}^T X_i(t)\right], \quad \forall w_i \in \mathcal{W},$$

$$Reg_j(T) = T \cdot \underline{\boldsymbol{U}}(a_j) - \mathbb{E}\left[\sum_{t=1}^T X_j(t)\right], \quad \forall a_j \in \mathcal{A},$$

where $\bar{\boldsymbol{U}}(w_i)$ represents the utility from the worker-optimal stable matching and $\underline{\boldsymbol{U}}(a_j)$ denotes the utility from the job-pessimal stable matching.

Previous work [60] studied regret minimization for both sides of the market in a setting without ties, adopting the regret definitions above. Zhang and Fang [60] establishes an $\mathcal{O}\left(K \log T/\Delta^2\right)$ regret bound for every agent, measured against their respective benchmark (worker-optimal and

job-pessimal). Our algorithm directly generalizes to the same setting and matches the theoretical guarantees of Zhang and Fang [60] when there are no ties. This is because the initial exploration phase will find the strict preference list for both sides simultaneously w.h.p. and thus, after committing, the resulting Gale-Shapley output will be the worker-optimal / job-pessimal stable matching.

In contrast, introducing ties to the market significantly complicates the analysis. The set of stable matchings expands substantially, and no single matching simultaneously satisfies the benchmarks defined for both sides. To extend our results to markets with ties, we propose using the Optimal Stable Share (OSS) as the benchmark for workers as defined in our paper. It is thus natural to introduce the equivalent Pessimal Stable Share (PSS) – the minimum utility a job can receive across all stable matchings. However, whether efficient algorithms can achieve sublinear regret for all agents in markets with ties remains open and would be interesting to explore.

