# OpenReview forum: "Stable Matching with Ties: Approximation Ratios and Learning"
_NeurIPS.cc/2025/Conference — NeurIPS 2025 poster_

### Official Review · Reviewer_tEhw · 2025-06-24

**Clarity:** 3
**Significance:** 3
**Originality:** 3
**Rating:** 5
**Confidence:** 4

**Summary:**

The paper studies the problem of stable matching in a two-sided setting where one side has ties in their cardinal utilities, while the other side has well-separated ordinal preferences.

The authors begin by analyzing the problem in the known-utilities setting. A key distinction from the classical case without ties is that there is no single matching that is globally preferred by the agents on one side. To characterize matchings in this setting, they propose the Optimal Stable Share Ratio (OSS-ratio), defined as the worst-case utility ratio achieved by any worker, with the general objective of finding a distribution over matchings that minimizes this ratio. They show that distributions over (weakly) stable matchings can result in a linear OSS-ratio in the number of agents, while restricting to internally stable matchings allows for a sublinear OSS-ratio, achievable via a proposed approximation algorithm. They further study a relaxation based on ε-stability, showing that a modified version of their algorithm guarantees expected utility within a logarithmic factor of the optimal ε-stable share.

The authors also study the problem in an online setting, where one side of the market learns its preferences through repeated interactions and bandit feedback. They propose an algorithm that achieves optimal regret bounds when the utilities can be easily separated (i.e., large $\Delta_{\min}$). In the more challenging regime, where $\Delta$ is small or zero (as in the case of ties), the algorithm bounds the regret relative to a proposed approximation oracle. Importantly, they highlight a fundamental trade-off in the achievable regret between these two regimes.

**Questions:**

**Q1.** Similar to Weakness point 1, I would like to ask about the analysis regarding the opposite side of the market. While it is clear that one side has known preferences, it also makes sense to consider the regret of the other side, since the initial observation of stability is an equilibrium in the market. To what extent can the regret be extended to every agent in the market? What potential future work do you foresee from this perspective?

Below, I enumerate several parts that I found somewhat confusing and believe would benefit from clarification.

**Q2.** In line 330: "... implementing an approximation oracle guarantees sublinear α-approximation regret." I am a bit confused by the term "sublinear." Isn't the bound in equation (7) linear?

**Q3.** In lines 371-372, the arguments are a bit difficult to follow. Could you please elaborate further?

**Q4.** In lines 673-675, I found the arguments somewhat unclear. Could you clarify them in more detail?

**Ethical Concerns:**

["NO or VERY MINOR ethics concerns only"]

**Final Justification:**

I would like to retain my positive score and recommend accepting the paper. The setting of matching with ties is both challenging and underexplored, and this work provides novel regret formulations, algorithms, and theoretical insights. The contribution is methodologically sound, and I did not identify any significant issues in the technical content. I believe it will be a useful addition to the literature.

**Limitations:**

yes

**Quality:**

4

**Strengths And Weaknesses:**

###  Strengths

1. **Clarity and Accessibility**: The paper is well written and easy to follow. It provides a comprehensive overview of the literature, using concrete examples that make the content accessible even to readers unfamiliar with the topic.

2. **Relevant and Practical Setting**: The focus on stable matching with ties is both theoretically interesting and practically relevant. The authors also consider the online learning setting, offering meaningful results for cases.

3. **Theoretical Contributions**: The paper presents a rich set of theoretical results with proofs that are clear and easy to read. I did not identify any errors in the proofs.

4. **Potential for Future Work**: The theoretical framework introduced in the paper can motivate further research, as discussed in the conclusion. Additionally, I find their approach to defining regret potentially interesting, especially in cases where a stable matching does not exist in the market.

### Weaknesses
Below, I enumerate several minor weaknesses of the paper.

1. The results of section 5.2 are focused on one side of the market (the workers). While this type of analysis is similar to prior work [39,40], I believe it would be beneficial to include an analysis of the regret on the other side of the market as well, since stability is a notion of equilibrium in the market that concerns both sides.

2.  While the paper is very well written, some parts/statements are difficult to follow (see also Question Q2). These could benefit from further clarification or restructuring.

3.  The paper does not include any experimental study or empirical comparison of methods. As the theoretical approach is well motivated I do not consider empirical evaluation necessary, a discussion of potential directions for future empirical work could enhance the paper and help connect the theoretical contributions to practical applications.

---

> ### Author Rebuttal · Authors · 2025-07-29
>
> We thank reviewer tEhw for the valuable comments. We provide response point by point below.
>
> - **Regret Analysis regarding the Other Side of the Market**
>
> The canonical result in two-sided matching markets without ties establishes that a worker-optimal stable matching is necessarily job-pessimal, demonstrating the fundamental impossibility of simultaneously achieving optimality for both sides of the market. Consequently, a natural approach to regret minimization in this setting involves considering both worker-optimal stable regret and job-pessimal stable regret as benchmarks. Previous work [1] has investigated this scenario, establishing an $O(K \log T / \Delta^2)$ regret bound relative to these respective benchmarks.
> In terms of our algorithm, if there are no ties, i.e., there is a minimal gap on both sides, our results immediately generalize to unknown job preferences and would guarantee the pessimal regret also for jobs.
>
> The introduction of ties significantly complicates the analysis, as the set of stable matchings expands substantially and loses the lattice structure present in settings with strict preferences. In particular, no single matching simultaneously achieves the benchmark utility for all agents in the market. One potential approach would be to define a Pessimal Stable Share (PSS) for jobs, analogous to our Optimal Stable Share (OSS) for workers, and investigate corresponding regret guarantees. While our work represents the first treatment of optimal stable regret in bandit learning for matching markets with ties, we follow prevailing research conventions by focusing primarily on the worker side. We regard the extension of our analysis to the job side as an important direction for future research.
>
> - **Clarification of the Paper**
>
> We thank the reviewer for the careful reading and we would incoporate the following explanations in the final version of the paper.
>
> **Q2**: ``Implementing an approximation oracle guarantees sublinear $\alpha$-approximation regret''
>
> In our proposed ETCO algorithm, $T_0$ is a hyperparameter. By setting it to an appropriate order relative to $T$, we could achieve an $\alpha$-approximation regret that is sublinear in $T$. In particular, we provide two choices of $T_0$ and the corresponding regret bounds in Corollary 1 and 2.
>
> **Q3**: Clarification for line 371-372: ``We would suffer linear regret for $\Delta_{\min} \in [\tilde{\Omega}(T^{-1/2}), \tilde{O}(T^{-1/3})]$, while keeping exploring till $\tilde{O}(T)$ we can have sublinear regret for this range.''
>
> From a statistical learning perspective, when the minimum preference gap $\Delta_{\min}$ is known to satisfies $\Delta_{\min} = \tilde{\Omega}(T^{-1/2})$, then $\tilde{O}(T)$ rounds of exploration suffice to reliably estimate worker preferences, and achieve sublinear regret under the definition in Eq.(3).
> However, in our setting where $\Delta_{\min}$ is unknown, the choice of exploration length $T_0$ introduces a fundamental trade-off between the two regret objectives in Eq.(3) and Eq.(4).
> By setting $T_0 = T^{2/3}(K \ln T)^{1/3}$, we cannot guarantee detection of instances where $\Delta_{\min}$ falls in the intermediate regime $[\tilde{\Omega}(T^{-1/2}), \tilde{O}(T^{-1/3})]$. For these cases, we must resort to the approximation oracle during exploitation. Cruicially, since the oracle's solution differs by a constant factor from the Gale-Shapley optimal, each exploitation round incurs constant regret when measured against Eq.(3), resulting in an overall linear regret.
>
> **Q4**: Clarification for line 673-675: ``Given that $U^*(w) = 1$ for any $w$, the ratio $R_{\mathcal{M}}$ is equal to $\min_D \max_w \frac{1}{U_D(w)}$ for these instances, which is no less than the ratio between the number of left nodes and the number of connected right nodes.''
>
> By substituting $U^*(w) = 1$ into $R_{\mathcal{M}}$, we get $\min_D \max_w \frac{1}{U_D(w)}$. Then, proving a lower bound on this quantity is equivalent to establishing an upper bound on $\max_D \min_w U_D(w)$. In particular, we have $\max_D \min_w U_D(w) \leq \max_D \frac{\sum_w U_D(w)}{N}$, where $N$ is the number of left nodes. Now notice that $\sum_w U_D(w)$ is the expected size of the matching under distribution $D$ (since the utility is $1$ when a worker is matched and $0$ otherwise). Since there are $K$ jobs, the size of any matching is bounded by $K$, and we have $\sum_w U_D(w) \leq K$, which results in the claim.
>
> - **Empirical Comparison of the Methods**
>
> While empirical experiments are valuable, this paper focuses on introducing a novel metric and rigorously analyzing its theoretical properties in both offline and online settings. Notably, no existing algorithms provide theoretical guarantees for our proposed framework.
>
> Prior work has extensively studied bandit learning in matching markets without ties, but these methods fail entirely when ties are introduced, as established by theoretical results. In contrast, our algorithm not only addresses this gap but also naturally reduces to the SOTA method [2] in tie-free settings.
>
> That said, we fully acknowledge the reviewer's point regarding the importance of empirical validation, given the broad potential applications of our framework. To highlight this, we will include a discussion of promising directions for future experimental research in the paper's concluding section.
>
>
> [1] Zhang, YiRui, and Zhixuan Fang. "Decentralized two-sided bandit learning in matching market." The 40th Conference on Uncertainty in Artificial Intelligence. 2024.
>
> [2] Kong, Fang, and Shuai Li. "Player-optimal stable regret for bandit learning in matching markets." Proceedings of the 2023 Annual ACM-SIAM Symposium on Discrete Algorithms (SODA). Society for Industrial and Applied Mathematics, 2023.

---

> > ### Comment · Reviewer_tEhw · 2025-08-01
> > **Additional questions regarding Q1**
> >
> > Dear Authors, thank you for your response.
> >
> > Regarding Question **Q1**, I would like to point out that the regret term $\text{Reg}_i$, as defined in Eq.3, considers only one side of the market (workers i.e., $ i \in W $). As a result, this benchmark does not fully capture the motivation behind stability, which inherently involves both sides of the market.
> >
> > While I acknowledge that this is a standard approach for analysis, could you elaborate further on this point? For example:
> > - Is it possible to define or bound the regret for each agent on both sides (i.e., for both workers and jobs)?
> > - How would such a definition align with the stability objective?
> >
> > I found the rest of your response clear and satisfactory

---

> > > ### Author Response · Authors · 2025-08-01
> > > **Regret Analysis for Both Sides of the Market**
> > >
> > > We thank reviewer tEhw for the further response.
> > >
> > > - **Define and Bound the Regret for Both Sides in a Market Without Ties**
> > >
> > > Yes, it is indeed possible to define regret for both sides of the market. Consider a matching market without ties. For each worker $w_i \in W$, the regret can be defined as $Reg_i(T)=T \cdot \bar{U}(w_i) - E \left[\sum_{t = 1}^T X_i(t)\right]$, where $\bar{U}(w_i)$ represents the utility from the worker-optimal stable matching. Conversely, for every job $a_j \in A$, the regret is $Reg_j(T) = T \cdot \underline{U}(a_j) - E [\sum_{t = 1}^T X_j(t)]$, where $\underline{U}(a_j)$ denotes the utility from the job-pessimal stable matching (that is, the minimal utility obtained under some stable matching). This distinction in benchmarks -- optimal for workers and pessimal for jobs -- stems from a fundamental result in two-sided matching theory: the worker-optimal stable matching is necessarily job-pessimal, and no stable matching simultaneously maximizes utility for both sides.
> > >
> > > Previous work [1] studied regret minimization for both sides of the market in a setting without ties, adopting the regret definitions above. The authors established an $O(K \log T / \Delta^2)$ regret bound for every agent, measured against their respective benchmark (worker-optimal and job-pessimal stable matching). Our algorithm directly generalizes to the same setting and matches the theoretical guarantees of [1] when there are no ties. This is because the initial exploration phase will find the strict preference list for both sides simultaneously (w.h.p.), and thus, after commiting, the resulting GS output will be the worker-optimal/job-pessimal stable matching.
> > >
> > > - **The Regret Definition Align with the Stability Objective**
> > >
> > > In a matching market without ties, under the given regret definition for both sides, if an algorithm achieves sublinear regret for every agent in the market, the sequence of implemented matchings must converge to the unique worker-optimal (and job-pessimal) stable matching.
> > >
> > > - **Extension to a Market with Ties**
> > >
> > > In contrast, introducing ties to the market significantly complicates the analysis. The set of stable matchings expands substantially, and no single matching simultaneously satisfies the benchmarks defined for both sides. To extend our results to markets with ties, we propose using the Optimal Stable Share (OSS) as the benchmark for workers (as defined in our paper). It is thus natural to introduce the equivalent Pessimal Stable Share (PSS) -- the minimum utility a job can receive across all stable matchings. However, whether efficient algorithms can achieve sublinear regret for all agents in markets with ties remains an open question, which we leave for future work.
> > >
> > > [1] Zhang, YiRui, and Zhixuan Fang. "Decentralized two-sided bandit learning in matching market." The 40th Conference on Uncertainty in Artificial Intelligence. 2024.

---

> > > > ### Comment · Reviewer_tEhw · 2025-08-04
> > > >
> > > > Thank you for your clarification. I suggest including the comments regarding regret in the appendix of your work to avoid any confusion.

---

> > > > > ### Author Response · Authors · 2025-08-04
> > > > >
> > > > > Thanks for your suggestion. We will include the above explanation regarding the regret analysis for both sides of the market in the appendix of the final version.

---

### Official Review · Reviewer_SsEk · 2025-07-01

**Clarity:** 4
**Significance:** 4
**Originality:** 4
**Rating:** 5
**Confidence:** 4

**Summary:**

This paper studies the problem of matching workers to jobs in a market where each participant has a preference ordering over the other side. It focuses on a key question: Is it possible to approximate a stable solution in markets with tied preferences that guarantees every worker a fair, minimum level of satisfaction? To address this, the paper defines a worker’s **optimal stable share (OSS)** as the maximum utility the worker can attain across all stable matchings. It then introduces a fairness metric called the **OSS-ratio**, which measures the fraction of a worker’s OSS that is guaranteed under a minimizing distribution over a given set of matchings. The considered sets include all matchings, all stable matchings, and all internally stable matchings.

The paper first shows that when restricted to distributions over stable matchings, the OSS-ratio is lower bounded by $\Omega(N)$, where $N$ is the number of workers. It then demonstrates that restricting to internally stable matchings yields an OSS-ratio upper bound of $\mathcal{O}(\log N)$, and provides an algorithm that achieves this bound while also being dominant-strategy incentive compatible. This bound is tight, as even without such restrictions, the OSS-ratio is $\Omega(\log N)$. Moreover, the positive results are robust to uncertainty in utilities.

The paper also explores a bandit learning framework. It introduces a key instance-dependent quantity called the **minimum preference gap** and provides a corresponding regret upper bound. Additionally, it defines a **relevant utility gap** and establishes a trade-off between regret and approximation regret.

**Questions:**

1. Is there a set of matchings that smoothly interpolates the OSS-ratio from $\mathcal{O}(\log N)$ to $\mathcal{O}(N)$? More specifically, for any $p > 1$, can one identify a set of matchings such that the OSS-ratio is $\Theta(N^{1/p})$?
2. Is the upper bound in Theorem 7 tight, or is it possible to improve the dependence on $\Delta_{\min}$?

**Ethical Concerns:**

["NO or VERY MINOR ethics concerns only"]

**Final Justification:**

The paper has novel aspects to it (see the Strengths section for more details). Moreover, the authors have responded to my questions in a satisfactory manner. Hence, I am retaining my current rating of '5: Accept'.

**Limitations:**

No such limitation

**Quality:**

4

**Strengths And Weaknesses:**

The main novelty of this work lies in the introduction of the fairness metric OSS-ratio, along with the accompanying positive and negative results. This is a valuable contribution to the game theory and fairness and has the potential to open a new line of research in this space. The proposed algorithm incorporates novel components such as the duplication index, which may offer fresh insights into stable matching problems. Additionally, this work is significant for the bandit learning community, as it introduces a new and compelling problem that could inspire a new subfield within the area.

---

> ### Author Rebuttal · Authors · 2025-07-29
>
> We thank reviewer SsEk for the valuable comments. We provide response point by point below.
>
> - **Set of Matchings that Interpolates the OSS-ratio from $O(\log N)$ to $O(N)$**
>
> We thank the reviewer for raising this insightful question. While this topic falls outside the scope of our current paper, we agree it represents an interesting direction for future research. Although we do not yet have a definitive answer, one promising approach would be to investigate constraints on the number of unmatched workers involved in a blocking pair: with no such workers, the OSS can scale linearly with $N$, while with $N$ potential blocking workers, we recover the $O(\log N)$ bound.
>
> - **Tightness of the Upper Bound in Theorem 7**
>
> Our upper bound in Theorem 7 matches the lower bound established by [1] in terms of both $T$ and $\Delta_{\min}$ when the minimum preference gap $\Delta_{\min}$ is sufficiently large.
>
> For preference profiles that may contain (statistical) ties, by setting the exploration length $T_0$ as $\tilde{O}(T^{2/3})$, our upper bound achieves the same dependence on $T$ as the lower bound in Theorem 8, though with $\Delta_{rel}$ replacing $\Delta_{\min}$. We emphasize that this represents a computational limitation rather than a fundamental statistical gap. As noted in footnote 2 in page 8, given either unlimited computational resources or an oracle capable of determining whether the uncertainty set contains a unique worker-optimal stable matching, we could close this gap and provide an upper bound that depends on $\Delta_{rel}$ instead of$\Delta_{\min}$.
>
> [1] Sankararaman, Abishek, Soumya Basu, and Karthik Abinav Sankararaman. "Dominate or delete: Decentralized competing bandits in serial dictatorship." International Conference on Artificial Intelligence and Statistics. PMLR, 2021.

---

> > ### Comment · Reviewer_SsEk · 2025-08-01
> >
> > Thank you for the rebuttal. I am satisfied with your response and will be retaining my current score.

---

### Official Review · Reviewer_yExD · 2025-07-02

**Clarity:** 3
**Significance:** 2
**Originality:** 3
**Rating:** 3
**Confidence:** 2

**Summary:**

The authors study the stable matching problem where one side (workers) of the market has ties in their preference lists over the other side (jobs). The workers have cardinal utilities over the jobs, and they consider the problem of optimizing for a new notion they define called the optimal stable share (OSS) ratio. This measures the ratio of the utility obtained by a worker compared to the best utility they could have obtained in a stable matching. They extend their results into two other settings, one where the OSS is defined with respect to \epsilon-stable matchings, and an online bandit setting where the utilities of the workers are not known but learned.

**Questions:**

Please respond to the concerns raised in the strengths and weaknesses section.

**Ethical Concerns:**

["NO or VERY MINOR ethics concerns only"]

**Final Justification:**

The paper will benefit from being focused more towards the learning theory results. The offline results do not seem very interesting.

**Limitations:**

yes

**Quality:**

2

**Strengths And Weaknesses:**

Overall, they define a notion of fairness for workers and somewhat fully investigate how much this can be achieved. While this is interesting as a theoretical exercise, I have concerns about any kind of practical viability, at least for sections 3, and 4.

Theorem 1 seems like a natural outcome of combining cardinal utilities with a notion more appropriate for ordinal utilities like stability. Theorem 2 and 3 are more interesting, but I think this is where my reservations with their notion of OSS lies. It seems far too restrictive, and leaving top workers unassigned with probability 1-1/log N appears to be necessary to get an optimal matching with rest to OSS. I can't think of a scenario where the jobs would prefer to do this, rather than formulate a different notion of fairness. I also don't see why it makes sense to optimize for the OSS when the ultimate distribution is over unstable matchings. Internal stability seems too weak a criteria for the output matchings.

Theorems 7, 8 seem more technically interesting, although I don't have enough of a learning theory background to comment on their novelty. My concerns persist about the applicability of OSS in this setting as well. Why should this be the metric of choice here?

---

> ### Author Rebuttal · Authors · 2025-07-29
>
> We thank reviewer yExD for the valuable comments. We provide our response point by point below.
>
> - **Leaving Top-workers Unassigned with Probability $1 - 1/\log N$**
>
> In the proposed algorithm, each worker is indeed assigned a job with a probability of $1 / \log N$. This is sufficient to prove a tight upper bound in Theorem 3 which matches the lower bound derived Theorem 2. However, some matchings in the support only assign a subset of jobs. In practice, if some job $a$ is not allocated in a matching $\tilde{\mu}_j$, but is allocated to worker $w$ in $\tilde{\mu}_i$, we can give $a$ to $w$ (which are mostly top workers) in $\tilde{\mu}_j$ without breaking internal stability of $\tilde{\mu}_j$. This post-processing is a Pareto improvement of our solution, but it does not provide any improved OSS guarantee in the worst case.
>
> - **Optimize for the OSS over Internally Stable Matchings**
>
> By interpreting the distribution over matchings as a rotating schedule, where each matching in the support represents a daily assignment, we can view the restriction to internally stable matchings as imposing constraints on this schedule. Specifically, it ensures that no worker experiences justified envy toward another worker present on the same day. Theorem 1 demonstrates that no meaningful guarantees can be established when considering justified envy of the workers who are not present, which naturally motivates us to explore the set of internally stable matching.
>
> Further, we would like to emphasize that the distribution computed by Algorithm 1 is not only "ex-post" internally stable, it is also "ex-ante" (externally) stable, in the sense that no worker has justified envy towards any other worker's (randomized) allocation. This is related to the notion of fractional stable matching discussed in Appendix A. We will add this as a comment in the final version, but we believe that formally defining ``ex-ante'' stability would distract the reader.
>
> - **Applicability of OSS in the Bandit Learning Setting**
>
> The bandit learning setting provides a natural and strong motivation for defining the OSS-ratio. Recent years have witnessed a surge of research on bandit learning in matching markets, where the goal is to minimize worker-optimal stable regret as defined in Eq.(3), using OSS as the benchmark. However, existing regret bounds universally scale as $1/\Delta^2$, where $\Delta$ represents the minimum utility preference gap across all worker-job pairs. This reveals a fundamental limitation: current methods fail completely when statistical ties exist ($\Delta = O(1/\sqrt{T})$).
>
> To address this challenge, we must refine the regret definition and identify a tractable benchmark. Drawing inspiration from approximation regret in combinatorial bandits, we propose using an $\alpha$-approximation of the OSS as our benchmark, where $\alpha$ corresponds precisely to the reciprocal of the OSS-ratio from the offline setting. Our approach first solves the offline problem, yielding an algorithm that serves as an offline oracle for bandit learning. This innovation enables meaningful regret guarantees even with potential statistical ties. Notably, by leveraging the OSS-ratio, we establish the first problem-independent regret upper bound -- a result previously believed impossible, as demonstrated by [1].
>
> [1] Liu, Lydia T., Horia Mania, and Michael Jordan. "Competing bandits in matching markets." International Conference on Artificial Intelligence and Statistics. PMLR, 2020.

---

> > ### Comment · Reviewer_yExD · 2025-08-08
> >
> > Thank you for your response. My suggestion would be to rewrite the introduction and abstract significantly so as to put the learning theory results front and center. In my opinion, the offline parts (which is what I focused on given my non-learning theory background) are not very impressive since all that was done was defining a new fairness metric that you showed was hard to achieve in general and showed that the worst case guarantee can be recovered. I still don't see any good motivation for this fairness metric and result in the offline setting.

---

> > > ### Author Response · Authors · 2025-08-08
> > >
> > > Thank you for your further response. We sincerely appreciate your thoughtful feedback and the time you have taken to review our work. While we deeply value your insights and will carefully consider them in future revisions, we respectfully disagree with some of the suggestions regarding the paper's organization.
> > >
> > > **We believe the current structure offers unity and completeness in understanding the OSS-ratio in stable matching with ties** -- we first characterize the bounds for the approximation ratio in the offline setting, and then move to regret analysis in bandit learning, using the offline results as a building block. Moreover, **we believe the offline results are well-motivated** (as also noted by reviewers wBg1 and tEhw) and **established key fairness trade-offs in stable matching with ties**, which reviewer SsEk highlighted as a valuable contribution to game theory and fairness research.
> > >
> > > That said, we fully acknowledge your concerns regarding the paper's presentation. If other reviewers and the AC share your perspective, we would be happy to revise the manuscript to further emphasize the online learning part.

---

### Official Review · Reviewer_wBg1 · 2025-07-03

**Clarity:** 2
**Significance:** 2
**Originality:** 2
**Rating:** 3
**Confidence:** 3

**Summary:**

The paper studies a stable matching problem where there are ties between the preferences. In particular, there is a set of workers and a set of jobs, and both of these sets have preferences over the other set. These preferences might have ties, e.g. an agent-worker might be indifferent between 2 jobs. The goal here is to produce a matching where it is guaranteed that each participant will acquire a utility that is above a certain threshold. The authors define the optimal stable share (OSS), i.e., the  maximum utility that can be achieved at a stable matching, as this threshold, and prove the following results:
1)	The authors design an algorithm that uses distributions over (possibly non-stable) matchings and show that an O(logn), where n is the number of workers, tight approximation to OSS ratio can be achieved
2)	They show that this result extends even when there is (bounded) uncertainty about the utilities of the participants
3)	They further extend these results to a bandit learning setting where utilities are only observed for matched pairs

**Questions:**

N/A

**Ethical Concerns:**

["NO or VERY MINOR ethics concerns only"]

**Final Justification:**

I confirm that I have read the rebuttal and the other reviews. My opinion and score remain the same.

**Limitations:**

yes

**Quality:**

3

**Strengths And Weaknesses:**

This paper studies a well-motivated and interesting problem and provides a nice collection of results. From a technical point of view, the paper is non-trivial and presents results that are technically involved. As far as I checked, these results appear to be also sound and correct. There is also some conceptual contribution due to the notion of OSS that I would say that I found very interesting in general.

On the other hand, the writing is not that clear and at least for me the paper was not that easy to follow. There are some structural problems, especially in the introductory sections that I believe should be revised in a way that demonstrates better what this work studies. The same goes for the contributions section. An example is that the paper talks about distributions of matchings, but initially it was not clear for me where these distributions come from, or how they are being used. From a technical perspective, as I said, the paper is involved, but I am not so sure about what new it adds in terms of techniques.

Overall, I would say that this is an interesting paper, but also one that would benefit from a revision so that the model and the overall contribution is more clear.

---

> ### Author Rebuttal · Authors · 2025-07-29
>
> We thank reviewer wBg1 for the valuable comments. We provide our response point by point below.
>
> - **Distributions over Matchings**
>
> In stable matching without ties, a worker-optimal stable matching -- one that simultaneously maximizes the utility for all workers -- always exists. However, when preferences admit ties, this strong property no longer holds, as different workers may favor different stable matchings.
> To balance competing interests, we consider distributions over matchings.
> In our setting, where a company allocates jobs to workers, the allocation could be done in a time-sharing scenario. For example, if workers are employed five days a week, the company could implement a rotating schedule: each day's assignment corresponds to an integral matching. This could be defined as a distribution over matchings where the support contains the daily assignments and the workers' satisfiability could be measured by the matching utility in expectation. Importantly, randomizing over matchings has been a standard approach in fair resource allocation [1-3], where previous work have shown the equivalence with fractional matchings, thanks to Birkhoff-von Neumann (BvN) theorem [4, 5].
>
> Due to space constraints, we initially deferred the above explanations to Appendix A. We thank the reviewer for their valuable feedback -- we agree that integrating this content into the main text will enhance the clarity of our conceptual contribution, and we intend to use the extra page to give more details on it.
>
> - **Techniques Involved and Developed**
>
> The upper bound on the approximation ratio is the first key technical contribution of our paper. We establish this result via three main steps: 1) Introducing a novel component -- the duplication index -- into the algorithm design; 2) Constructing a directed forest where edges encode conflicts between workers competing for the same job copies across different matchings; 3) Leveraging the tree structure and stability constraints to derive the upper bound inductively.
>
> In the bandit learning setting, the primary technical challenge and key contribution lie in the lower bound proof. To establish this result, we carefully construct two instances with 4 workers and 4 jobs, where the utility matrices differ in only one critical entry that determines whether meaningful ties exist. This construction reveals how ties in one worker's preferences propagate to affect other workers' regret. Furthermore, we employ an information-theoretic argument to demonstrate that the algorithm must sample this critical entry sufficiently often to avoid incurring linear regret.
> To our knowledge, we are the first to provably show a tradeoff between standard regret and approximation regret in bandit settings.
>
> To clarify our technical contributions, we will add a paragraph that highlights our proof sketches to the introduction in the final version.
>
> - **Improvement in the Writing**
>
> We will incorporate the above discussions and revise the introduction section to further clarify our motivation and contributions in the final version of the paper. Additionally, we are happy to provide further clarification on any other aspects of the paper, as detailed in our response to reviewer tEhw. If there are additional clarity issues that we did not address, we would appreciate it if you could let us know so we could also incorporate these improvements into our final version.
>
> [1] Karp, Richard M., Umesh V. Vazirani, and Vijay V. Vazirani. "An optimal algorithm for on-line bipartite matching." Proceedings of the twenty-second annual ACM symposium on Theory of computing. 1990.
>
> [2] Halpern, Daniel, and Nisarg Shah. "Fair and efficient resource allocation with partial information." arxiv preprint arxiv:2105.10064 (2021).
>
> [3] Benade, Gerdus, et al. "Fair and efficient online allocations." Operations Research 72.4 (2024): 1438-1452.
>
> [4] Birkhoff, Garrett. "Three observations on linear algebra." Univ. Nac. Tacuman, Rev. Ser. A 5 (1946): 147-151.
>
> [5] Von Neumann, John. "A CERTAIN ZERO-SUM TWO-PERSON GAME EQUIVALENT TO THE OPTIMAL ASSIGNMENT PROBLEM¹." Contributions to the Theory of Games 24 (1950): 5.

---

### Note · Authors · 2025-08-16

We are deeply grateful to the Reviewers and the AC for their constructive comments and participation in the review process.

Our paper studies stable matching with ties, framing the problem as assigning workers to jobs where worker preferences over jobs may admit ties. We introduce the OSS-ratio, a novel fairness metric that guarantees workers a minimal fraction of their optimal stable share. **Our complete characterization spans offline and online settings, matching algorithmic upper bounds with fundamental lower bounds through interdisciplinary techniques.**

Our main contributions are as follows:

1. **Novel fairness metric in stable matching with ties**: We introduce the OSS-ratio, a natural and well-motivated fairness metric that captures the fundamental trade-off in stable matching with ties.

2. **Tight OSS-ratio bounds and offline approximation oracle**: For distributions over stable matchings, we prove an $\Omega(N)$ lower bound, improved to $\Omega(\log N)$ when considering all matchings. We match this with an efficient algorithm achieving $O(\log N)$ via distributions over internally stable matchings.

3. **Bandit learning with approximation regret**: We define a new approximation regret for the bandit setting, where the approximation ratio is the reciprocal of the OSS-ratio. This enables strong regret guarantees even with statistical ties -- overcoming a key limitation of existing methods.

4. **Tight regret bounds**: Our adaptive algorithm achieves optimal regret in both tie-free settings (matching the lower bound from Sankararaman et al. (2021)) and settings with statistical ties (establishing novel matching bounds). This yields the first provable trade-off between standard and approximation regret.

5. **Key technical contributions**: We develop: (1) a duplication index and conflict forest for a tight offline upper bound via inductive stability arguments; (2) an online lower bound via paired instances differing in one critical utility entry, exposing regret propagation through ties with an information-theoretic sampling requirement.

To address the reviewers' concerns, we will clarify our conceptual and technical contributions through revisions as outlined in our rebuttal and discussion.

We believe that our results contribute to fairness in stable matching with ties, connecting insights from economics/game theory and online learning while paving the way for future research.

Thanks again for the time and effort for the reviewers and the AC.

---

### Decision · Program_Chairs · 2025-09-17

**Decision:**

Accept (poster)

**Comment:**

The paper introduces the Optimal Stable Share (OSS)-ratio, a new fairness metric for stable matching markets with ties, and provides tight approximation and regret bounds in both offline and bandit learning settings. Reviewers agree that the work tackles an important and underexplored problem at the intersection of economics, game theory, and online learning. Key strengths include introducing the OSS metric, tight technical results for both distributions over stable matchings and bandit learning, and a comprehensive theoretical framework potentially inspiring future research. However, concerns remain about the clarity of presentation, the practical motivation for the OSS metric, and the lack of broader consideration of both sides in the market. Notably, reviewers differ on whether to emphasize the offline or online (bandit) contributions. Given the novelty and technical depth recognized by multiple reviewers, especially for the introduction of a tractable fairness metric and rigorous learning results, the paper is recommended for acceptance. The authors are strongly recommended to clarify key motivations, update structural organization as suggested, and discuss more empirical and multi-sided implications in the camera ready (as recommended by the reviewers).